# AeroMix v1.0.1: a Python package for modeling aerosol optical properties and mixing states

Sam P. Raj [1], Puna Ram Sinha [1], Rohit Srivastava [2], Srinivas Bikkina [3], Damu Bala Subrahamanyam [4]

[1]Department of Earth and Space Sciences, Indian Institute of Space Science and Technology Thiruvananthapuram, Kerala, 695547, India

[2] National Centre for Polar and Ocean Research, Ministry of Earth Sciences, Govt. of India, Vasco-da-Gama, Goa, 403804, India

[3] Birbal Sahni Institute of Palaeosciences, 53, University Road, Lucknow, 226007, India

[4] Space Physics Laboratory, Vikram Sarabhai Space Centre, Thiruvananthapuram, Kerala, 695022, India

*Correspondence to*: P. R. Sinha, (prs@iist.ac.in)

**Abstract.** Assessing aerosol mixing states, which primarily depend on aerosol chemical compositions, is indispensable to estimate aerosol direct and indirect effects. The limitations in the direct measurements of aerosol chemical composition and mixing states necessitate modeling approaches to infer the aerosol mixing states. The Optical Properties of Aerosols and Clouds (OPAC) model has been extensively utilized to construct optically equivalent aerosol chemical compositions from measured aerosol optical properties using Mie inversion. However, the representation of real atmospheric aerosol mixing scenarios in OPAC has perennially been challenged by the exclusive assumption of external mixing. A Python successor to the aerosol module of the OPAC model is developed, named 'AeroMix,' with novel capabilities to 1) model externally and core-shell mixed aerosols, 2) simulate optical properties of aerosol mixtures constituted by any number of aerosol components, and 3) define aerosol composition and relative humidity in up to six vertical layers. Designed as a versatile open-source aerosol optical model framework, AeroMix is tailored for sophisticated inversion algorithms aimed at modeling aerosol mixing states and also their physical and chemical properties. AeroMix's performance is demonstrated by modeling the probable aerosol mixing states over Kanpur (urban), and the Bay of Bengal (marine) in South Asia. The modeled mixing states are consistent with independent measurements using single-particle soot photometer (SP2) and transmission electron microscopy (TEM), substantiating the potential capability of AeroMix to model complex aerosol mixing scenarios involving multiple internally mixed components in diverse environments. This work contributes a valuable tool for modeling aerosol mixing states to assess their impact on cloud nucleating properties and radiation budget.

## 1. Introduction

Various sources of aerosol particles and their multi-scale dynamic nature in the atmosphere constitute a complex mixture of externally and internally mixed aerosol components, which are highly variable spatially and temporally (Ching et al., 2019; Riemer et al., 2019). Knowledge of the size-resolved aerosol chemical composition, size distribution, and mixing state is

required to predict cloud condensation nuclei (CCN) concentrations for freshly emitted and aged aerosols (McFiggans et al., 2006; Ervens et al., 2010; Farmer et al., 2015). Thus, the impact of aerosol mixing state and chemical composition on the activation of CCN needs to be determined and understood to the extent that this is importantly represented in global climate
models (Ghan and Schwartz, 2007). However, in situ measurements of aerosol chemical compositions and mixing states are sparse due to the complexities associated with the measurement techniques (Riemer et al., 2019). The standard aerosol model outlined in Optical Properties of Aerosols and Clouds (OPAC) (Hess et al., 1998; Koepke et al., 2015) has been widely adopted to estimate the probable aerosol mixing state from the measured aerosol optical properties through Mie inversion. However, this is not the optimal approach to assess the aerosol composition and mixing state for a specific location, but this is the viable
and practical option when direct measurements of aerosol chemical composition are unavailable.

The OPAC model has significantly contributed to aerosol research by providing reliable simulations of the optical properties of different aerosol mixtures necessary to estimate their radiative effects. It can model user-defined aerosol mixtures by mixing up to seven aerosol components. Despite the experimental evidence that the aerosol mixing state lies between a purely externally mixed state and a purely internally mixed state (Healy et al., 2014; Li et al., 2016; Ye et al., 2018; Riemer et al.,
2019), the standard FORTRAN-based OPAC model considers the external mixing of aerosols alone and cannot treat complex aerosol internal mixing states. Most current climate models also assume entirely externally or internally mixed aerosols, resulting in an error in modeled optical, hygroscopic, CCN, and cloud properties (Stevens and Dastoor, 2019). Additionally, restricting the number of components constituting an aerosol mixture further limits the modeling of complex aerosol mixtures using OPAC.

Several attempts have been made to incorporate internal mixing in OPAC by modifying the predefined components with the optical properties of internally mixed aerosols (Chandra et al., 2004; Dey et al., 2008; Ramachandran and Srivastava, 2016; Srivastava et al., 2016, Srivastava et al., 2018). In these studies, the mixing state of aerosols using the OPAC model was modeled by iteratively comparing modeled optical parameters such as aerosol optical depth (AOD), single scattering albedo (SSA), and asymmetry parameter ($g$) with measured ones for different mixing scenarios until their values converge within the
observational error. Transmission electron microscopy (TEM) observations have shown the presence of multiple combinations of internally mixed aerosol components in a mixture (Li et al., 2016). However, the limitation in the number of aerosol components permissible to constitute an aerosol mixture in OPAC constrained the number of internally mixed components considered for each case in the above-referred studies. This limitation thus restricts the number of possible combinations of aerosol components to determine the probable mixing states.

In an effort to address this challenge, a Python-based package named 'AeroMix' for modeling aerosol optical properties and mixing states is developed. AeroMix enables the modeling of optical properties of complex aerosol mixtures consisting of any number of components in externally mixed and/or core-shell mixed states in up to six vertical layers. Furthermore, other internal mixing states can also be represented using appropriate mixing rules (Bohren and Huffman, 1998; Stevens and Dastoor, 2019). The scalability of the number of components in a mixture, the ability to model core-shell mixed aerosols, and the scope
of integration with other programs make AeroMix a versatile open-source tool for inversion algorithms aimed at modeling the

properties of aerosol mixtures, including aerosol chemical composition, and mixing states. For this purpose, AeroMix performance is demonstrated by determining the chemical composition and probable mixing states of aerosols from the measured aerosol optical properties using the Mie inversion technique over Kanpur, a representative urban location in the Indo-Gangetic Plains (IGP), and a marine environment of the Bay of Bengal (BoB) in South Asia. We initially present an overview of AeroMix in Sect. 2, followed by detailed methodology including the model setup and a comparison with the OPAC model in Sect. 3. The AeroMix modeled probable aerosol mixing states over Kanpur and BoB, along with a comparison of the results against independent measurements conducted by single-particle soot photometer (SP2) and TEM, is presented in Sect. 4.

## 2. Model overview

AeroMix is an open-source Python package developed to model the optical properties of aerosol mixtures, including AOD, SSA, asymmetry parameter, extinction coefficient ($\beta_{ext}$), scattering coefficient ($\beta_{sca}$), and absorption coefficient ($\beta_{abs}$) at sixty-one wavelengths ranging from 0.25 to 40 µm and eight relative humidity (RH) values, following Hess et al. (1998). The workflow of AeroMix for modeling the aerosol properties and assessing the probable mixing states is illustrated in Fig. 1. A methodology for modeling the probable aerosol mixing state using AeroMix is detailed in the following section.

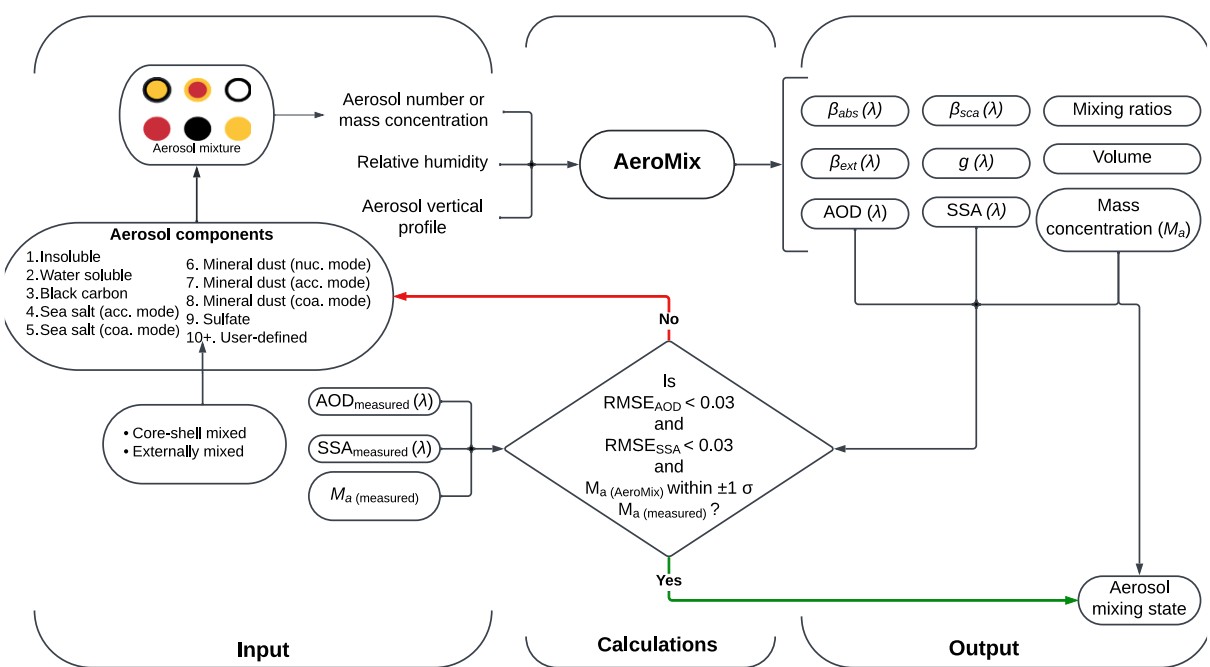

**Figure 1: Overview of the AeroMix workflow for modeling aerosol mixing states using the Mie inversion technique.**

## 2.1. Mixing of aerosol components

Aerosol mixtures in AeroMix can be defined in terms of the number or mass concentration of the constituent aerosol components in external (composed of single chemical species) and/or internal (composed of multiple chemical species as core-shell structure) mixed states, along with their vertical distribution and the vertical profile of RH. Optical properties of the complex aerosol mixing states are modeled by accounting for any number of both externally mixed particles and internally mixed particles, with no presumed chemical or physical interaction among the particles within the mixture (see Fig. 1).

### 2.1.1. Externally mixed aerosol components

Nine predefined externally mixed aerosol components in AeroMix include water-insoluble (IS), water-soluble (WS), black carbon (BC), accumulation and coarse modes of sea-salt (SSam and SScm), nucleation, accumulation and coarse modes of mineral dust (MDnm, MDam, and MDcm) and stratospheric sulfate (SU). These components are represented in terms of their lognormal size distribution parameters (geometric standard deviation ($\sigma$), mode radius ($r_m$) and the upper and lower limits of radius, $r_{min}$ and $r_{max}$), specific density ($\rho$), spectral refractive indices ($m$) and optical properties averaged for one particle ($\beta_{ext}$ ($\lambda$), $\beta_{sca}$ ($\lambda$), $\beta_{abs}$ ($\lambda$), SSA ($\lambda$) and $g$ ($\lambda$)) at sixty-one wavelengths from 0.25 to 40 µm and eight RH values, adopted from the Global Aerosol Data Set and OPAC database (Koepke et al., 1997; Hess et al., 1998; Koepke et al., 2015). A brief overview of the parameters used to describe the predefined aerosol components at dry state (RH = 0%) is given in Table 1. Detailed descriptions of the aerosol components, along with their size distribution parameters and spectral refractive indices, can be found elsewhere (Koepke et al., 1997; Hess et al., 1998; Koepke et al., 2015). The density of BC is set at 1.8 g cm$^{-3}$ based on observations, which differs from the value of 1.0 g cm$^{-3}$ given in OPAC (Bond et al., 2013; Kondo, 2015). The optical properties of each component, except for MDnm, MDam, and MDcm components, are calculated using the Mie theory (Mie, 1908). This calculation assumes that the particles are spherical in shape and follow a lognormal size distribution. The calculated optical properties are normalized to one particle cm$^{-3}$ and stored, which can then be scaled to any given number concentration. MDnm, MDam, and MDcm components are modeled using the T-Matrix Method (TMM) (Waterman, 1971; Mishchenko et al., 1999) to account for their non-sphericity (Koepke et al., 2015). In addition to the nine predefined aerosol components, AeroMix offers the flexibility to model any number of user-defined components. A new externally mixed aerosol component can be defined in AeroMix by its size distribution parameters, specific density, and spectral refractive indices described above.

**Table 1: Size distribution parameters, specific densities, and refractive indices of predefined aerosol components at dry state (RH = 0%).**

| Aerosol component | Constituting species | Mode radius (µm) | Geometric standard deviation | specific density (g cm$^{-3}$) | Refractive index at 0.5 µm |
|---|---|---|---|---|---|
| Insoluble (IS) | Soil dust, fly ash, and non-hygroscopic organic matter. | 0.471 | 2.51 | 2.0 | $1.53 + 8 \times 10^{-3}i$ |
| Water-soluble (WS) | $SO_4^{2-}, NO_3^-, NH_4^+$ and hygroscopic fraction of organic matter. | 0.0212 | 2.24 | 1.8 | $1.53 + 5 \times 10^{-3}i$ |
| Black carbon (BC) | Black carbon aerosols | 0.0118 | 2 | 1.8 | $1.75 + 4.5 \times 10^{-1}i$ |
| Sea-salt accumulation mode (SSam) | Sea salt aerosols | 0.209 | 2.03 | 2.2 | $1.5 + 1.55 \times 10^{-8}i$ |
| Sea-salt coarse mode (SSam) | | 1.75 | 2.03 | | |
| Mineral dust nucleation mode (MDnm) | Desert dust | 0.007 | 1.95 | 2.6 | $1.53 + 7.8 \times 10^{-3}i$ |
| Mineral dust accumulation mode (MDam) | | 0.39 | 2 | | |
| Mineral dust coarse mode (MDcm) | | 1.9 | 2.15 | | |
| Stratospheric sulfate (SU) | Sulfate aerosols from volcanic eruption. | 0.0695 | 2.03 | 1.7 | $1.431 + 1 \times 10^{-8}i$ |

110

The OPAC aerosol database is comprehensive, reliable, based on extensive observations, and widely employed in radiative transfer modeling, global climate modeling, and mixing state studies (Shettle et al., 1979; Deepak and Gerber, 1983; Koepke et al., 1997; Srivastava et al., 2016). Studies examining the sensitivity of refractive indices report negligible influences on modeled $\beta_{ext}$ (Ramachandran and Jayaraman, 2002; Srivastava et al., 2016). Further, Srivastava et al. (2016) investigated the sensitivity of BC mode radius on SSA and $\beta_{ext}$ for BC and sulfate in different mixing states, revealing differences of only up to 1.3%. Hence, the AeroMix modeled optical properties using the OPAC database are anticipated to be minimally affected by uncertainties in refractive indices and size distribution parameters. Along with the default database, AeroMix allows users to employ various datasets that characterize aerosols using the parameters described above. This flexibility not only enables users to choose datasets based on their preferences but also enhances AeroMix capability by incorporating more comprehensive datasets that consider the complex characteristics of aerosol particles, including morphology.

### 2.1.2. Internally mixed aerosol components

Core-shell mixed aerosol components can be defined in AeroMix by specifying the core and shell components with their core-to-shell radius ratio (CSR) or mass fractions of the species in a core-shell mixed particle. Optical properties of the core-shell mixed aerosol components are modeled by using PyMieScatt (Sumlin et al., 2018), a coated-sphere Python Mie calculation program based on the BHCOAT program (Bohren and Huffman, 1998). PyMieScatt takes the spectral refractive indices of the core and shell components, and the radius of the core ($r_c$) and shell ($r_s$) as input parameters to model the optical properties of the core-shell mixed components.

The $r_s$ for each particle is equivalent to the radius of the core-shell mixed particle and is assumed to follow the size distribution of the shell component (Srivastava et al., 2016), which is also supported by observations (Arimoto et al., 2006). The $r_c$ of each particle is calculated according to the CSR, expressed as the ratio of the $r_c$ to the $r_s$, and is given by,

$$CSR = \frac{r_c}{r_s} = \left(1 + \frac{M_s \rho_c}{M_c \rho_s}\right)^{(-1/3)},$$ ( 1 )

$M_c$ and $M_s$ are the mass contributions of core and shell components to the mixing, and $\rho_c$ and $\rho_s$ are the specific mass densities of core and shell components, respectively (Srivastava et al., 2016; Chandra et al., 2004). CSR can be specified in AeroMix either directly or in terms of the mass contribution of the core and shell components. The mass of the core-shell mixed components is calculated from their size distribution parameters and effective mass density ($\rho_{eff}$). The $\rho_{eff}$ of the core-shell mixed particle can be defined as,

$$\rho_{eff} = \frac{M_c + M_s}{V},$$ ( 2 )

where $V$ is the volume of the core-shell mixed particle. Since $M_c$, $M_s$, and $V$ vary along the particle size distribution as a function of $r_c$ and $r_s$, $\rho_{eff}$ needs to be defined in terms of parameters that remain fixed across the size distribution for modeling simplicity. For this, $M_c$ and $M_s$ can be written as,

$$M_c = \rho_c \frac{4}{3}\pi r_c^3 \text{ and } M_s = \rho_s \frac{4}{3}\pi (r_s^3 - r_c^3).$$ ( 3 )

Since, $CSR = \frac{r_c}{r_s}$,

$$\rho_{eff} = \rho_c CSR^3 + \rho_s(1 - CSR^3). \tag{4}$$

Equation (4) contains only the terms of the specific density (*ρ*) of core and shell components and CSR value, which remain the same across the size distribution.

## 2.2. Vertical profiles of aerosols and relative humidity

For total column AOD calculation, up to six vertically arranged layers can be defined to specify the vertical distribution of aerosols. Unlike OPAC, in AeroMix, aerosol concentrations at the layer base, aerosol profile type, and layer mean RH can be

defined separately for mixed layer, free troposphere, stratosphere, and elevated aerosol layers. In contrast, OPAC uses constant background extinction coefficient values and RH for the free troposphere and stratosphere. The vertical profile of aerosol concentration in each layer can be modeled as homogenous or as an exponential function given by,

$$N(h) = N(0)exp\left(-h/z\right), \tag{5}$$

where *N(0)* is the number concentration of aerosol at the layer bottom, *h* is the height from the layer bottom, and *z* is the scale

height representing the change in aerosol concentration with height (Hess et al., 1998). Standard exponential profiles for different aerosol types provided by Hess et al. (1998) can be utilized for locations lacking aerosol vertical profile measurements. Alternatively, it can be modeled using a cubic function (Eq. (6)) when aerosol vertical profile measurements are available.

$$N(h) = N(0)(ah^3 + bh^2 + ch + d), \tag{6}$$

where *a*, *b*, *c,* and *d* are the coefficients when $N(0) = 1$ (Russo et al., 2006). The default values of aerosol concentration, RH, and profile type for the free troposphere, stratosphere, and elevated mineral dust layer are adopted from Hess et al. (1998). The total column AOD is calculated by,

$$Total\ AOD = \sum_{n=1}^{6} \beta_{ext_n} \int_{h_{min_n}}^{h_{max_n}} N_n(h)dh \tag{7}$$

where $h_{min}$ and $h_{max}$ are the layer bottom and layer top height for each layer $n$.

The AeroMix package and detailed documentation are available online at www.github.com/sampr7/AeroMix (P Raj and Sinha, 2024a).

## 3. Modeling aerosol mixing state with AeroMix

The primary objective of AeroMix is to provide an open-source aerosol optical model framework tailored to support inversion

algorithms for modeling both aerosol mixing states and their physical and chemical characteristics. The probable aerosol

mixing states are modeled with AeroMix using the Mie inversion technique by iteratively comparing the multispectral measurements of aerosol optical properties with modeled ones for different mixing scenarios until they converge within the observational error (Fig. 1) (Chandra et al., 2004; Dey et al., 2008; Kaskaoutis et al., 2011; Ramachandran and Srivastava, 2016; Srivastava et al., 2016, Srivastava et al., 2018).

### 3.1. Comparison of AeroMix with OPAC

Initially, the AeroMix modeled aerosol properties were compared with OPAC for ten externally mixed cases given in OPAC, namely continental clean, continental average, continental polluted, urban, desert, maritime clean, maritime polluted, maritime tropical, Arctic, and Antarctic. The AeroMix computed aerosol mass concentrations ($M_a$) of all ten cases, as well as AOD, SSA, and asymmetry parameters, showed excellent agreement ($r = 0.99$, slope = ~ 1) with OPAC-derived ones (Figs. A1 and A2 in Appendix A).

### 3.2. Model setup

### 3.2.1. Study region and data

AeroMix performance was further assessed by determining the probable aerosol mixing states over two contrasting environments: Kanpur, India (26.513° N, 80.232° E, 123 m AMSL; urban) and the Bay of Bengal (11.99°- 20.61°N, 80.52°- 92.55° E; BoB; marine) representing diverse aerosol mixtures. Past endeavors to deduce intricate aerosol mixing states through Mie model inversion were hindered by the OPAC model's constraints on the number of plausible mixing state cases. The efficacy of this method relies on the accuracy of input parameters, encompassing AOD, SSA, aerosol vertical profile, RH, refractive index, size distribution, and mixing state assumptions. Therefore, collocated and concurrent measurements of quality-controlled spectral AODs, spectral SSAs, spectral asymmetry parameter, vertical profile of aerosols, aerosol chemical composition, mixed layer height (MLH), and RH were used for the first time over Kanpur and the BoB; in South Asia to model the probable mixing states of aerosols using AeroMix. These measurements were taken from January 2007 to December 2009 at Kanpur and during the Winter-Integrated Campaign for Aerosols, gases and Radiation Budget (W-ICARB) conducted from December 2008 to January 2009 over the BoB (Moorthy et al., 2010). The seasonal mean MLH and $\beta_{ext}$ profiles at 532 nm obtained from NASA Micro-Pulse Lidar Network (MPLNET) collected over Kanpur during the period May 2009 – November 2015 are utilized in this study. $\beta_{ext}$ profiles at 532 nm over the western-BoB (W-BoB) and northern-BoB (N-BoB) are obtained from Cloud-Aerosol Lidar with Orthogonal Polarization (CALIOP). A summary of various aerosol and meteorological datasets used in the present study is provided in Table 2. A detailed description of the datasets utilized in this study can be found in the reference therein and presented in Appendix B. The locations of Kanpur, W-BoB, N-BoB, and the W-ICARB cruise track, along with seasonal average surface wind directions, are presented in Fig. 2.

**Table 2: Summary of aerosol and meteorological data utilized in this study.**

| Kanpur | | | | | |
|---|---|---|---|---|---|
| **Parameter** | **Period** | **Resolution** | | | **Reference** |
| | | **Temporal** | **Spatial** | **Vertical** | |
| Aerosol optical depth (AOD) at 0.34, 0.38, 0.44, 0.5, 0.675, and 0.87 μm | January 2007 – December 2009 | Monthly | - | - | AERONET Dubovik and King, 2000; Holben et al., 2001 |
| Single scattering albedo (SSA) at 0.44, 0.675, 0.87, and 1.02 μm | | | | | |
| Asymmetry parameter ($g$) at 0.44, 0.675, 0.87, and 1.02 μm | | | | | |
| Aerosol extinction coefficient ($\beta_{ext}$) at 532 nm | May 2009 – November 2015 | 1 minute | - | 75 m | MPLNET Welton et al., 2001 |
| Aerosol chemical composition | January 2007 – March 2008 | 1-2 samples/ week | - | - | Ram et al. 2010a |
| Mixed layer height (MLH) | May 2009 – November 2015 | 1 minute | - | - | MPLNET Lewis et al., 2013 |
| Relative humidity (RH) | November 2007 – December 2009 | Hourly | - | - | MOSDAC |
| **Bay of Bengal** | | | | | |
| **Parameters** | **Period** | **Resolution** | | | **Reference** |
| | | **Temporal** | **Spatial** | **Vertical** | |
| Aerosol optical depth (AOD) at 0.38, 0.44, 0.5, 0.675, and 0.87 μm | 27 December 2008 – 9 January 2009 | 10 minutes | - | - | Kaskaoutis et al. 2011 |
| Aerosol extinction coefficient ($\beta_{ext}$) at 532 nm | December 2008 – January 2009 | Monthly | 2°×5° (lat×lon) | 60 m | |

| | | | | | CALIOP<br>Tackett et al., 2018 |
|---|---|---|---|---|---|
| Aerosol chemical composition | 27 December 2008 – 9 January 2009 | Daily | - | - | Srinivas et al. 2011 |
| Mixed layer height (MLH) | 27 December 2008 – 9 January 2009 | 2 launches/ day | - | - | Subrahamanyam et al. 2012 |
| Relative humidity (RH) | 27 December 2008 – 9 January 2009 | Daily | - | - | Sinha et al., 2011b |

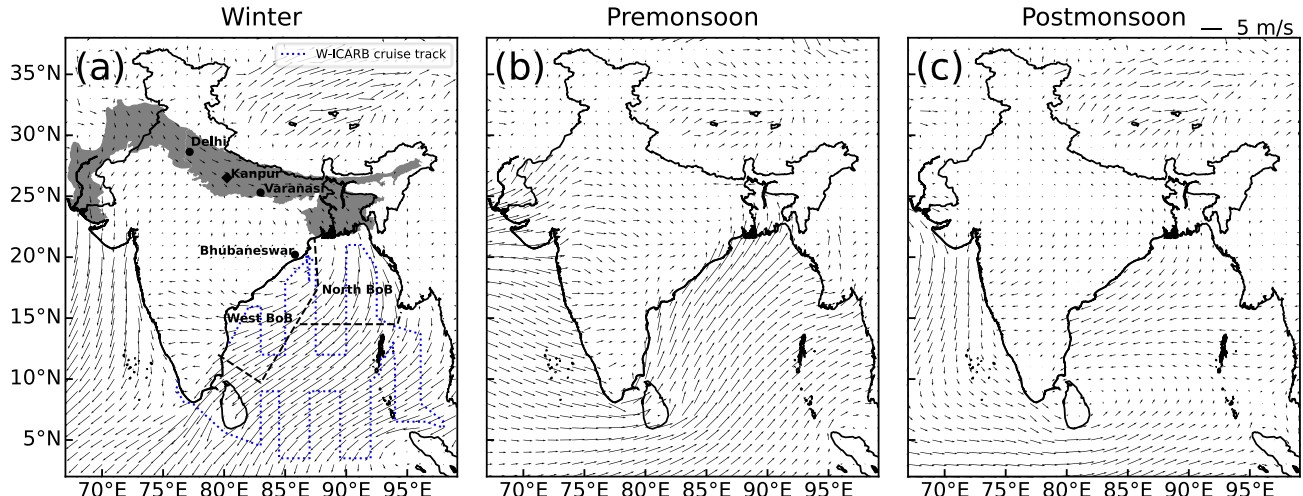

**Figure 2: a) Location of Kanpur (diamond) and the W-ICARB cruise track (dotted lines). Measurements of AOD, chemical composition, and RH over the West-BoB and North-BoB (demarcated with dashed lines) were utilized in this study. Additionally, the locations of Delhi, Varanasi, and Bhubaneswar (circles) are indicated, where mixing state measurements were obtained for comparison with the modeled mixing states. The shaded area marks the Indo-Gangetic Plains. Seasonal average surface winds over the region from ERA5 reanalysis are shown for a) winter, b) premonsoon and c) postmonsoon by vectors.**

### 3.2.2. Mixing state assumptions

The analysis encompasses eight aerosol components (IS, WS, BC, SSam, SScm, MDnm, MDam, and MDcm) in an externally mixed state and their combinations forming a core-shell mixed structure at various CSR values, ranging from 0.1 to 0.9 with an increment of 0.1. This results in a total of 305 distinct and plausible aerosol components under consideration. This excludes
combinations involving MD as the shell component and the homogenous internal mixing of these aerosol components, as their physical existence is rather unlikely in the atmosphere (Jacobson, 2000; Dey et al., 2008). The SU component is not explicitly considered, as the WS component accounts for anthropogenic sulfate. Different CSR values signify distinct mixing scenarios of core and shell components, each with varying mass fractions within a particle. For instance, a CSR of 0.1 indicates a thick coating, while a CSR of 0.9 suggests a thin coating.

Previous studies have assumed either the entire mass of the constituent aerosol components or a specific fraction of it to be involved in core-shell mixing (Chandra et al., 2004; Dey et al., 2008; Srivastava and Ramachandran, 2013). The latter assumption aligns more closely with observed mixing scenarios. Wherein, the entire mass of an aerosol component may not necessarily be in a core-shell mixed state with another. Instead, a fraction of it may be, while the rest can exist in an externally mixed state or be internally mixed with other aerosol components (Arimoto et al., 2006; Shamjad et al., 2016; Thamban et al.,
2017). However, both assumptions necessitate prior knowledge of the component-wise $M_a$ for computing the CSR value of core-shell mixed aerosols to determine their optical properties. In contrast, this study proposes allowing the variations of CSR values of core-shell mixed aerosols in the model. This offers a more flexible approach than relying on measured $M_a$ to assess the probable aerosol mixing state. Subsequently, the mass fraction of the core and shell components participating in the mixing can be calculated from the CSR value of the probable core-shell mixed components using Eq. (1).

The aerosol mixing state in this study is described based on the distribution of the mass of different aerosol components within the aerosol mixture. If each particle in the mixture is composed of a single aerosol component, the mixture is considered 100% externally mixed. Conversely, if all the particles are composed of two aerosol components as a core-shell structure, it is denoted as 100% core-shell mixed. The total mass of the particles composed of a single aerosol component contributes to the externally mixed aerosol mass fraction, while the total mass of particles with a core-shell structure contributes to the core-shell mixed
mass fraction. The mass fraction of an aerosol component in a core-shell mixed component is determined by calculating the ratio of its mass in that core-shell mixed component to the total mass of that component in the aerosol mixture. The masses of core and shell components in a core-shell mixed component are computed using Eq. (1). The combined mass of an aerosol component in various core-shell mixed particles and in an externally mixed state contributes to the total mass of that aerosol component in the mixture. The combined mass concentrations of SSam and SScm together are referred to as SS mass
concentration ($M_{SS}$), while MDnm, MDam, and MDcm collectively constitute the MD mass concentration ($M_{MD}$). Here onwards, IS, WS, and BC mass concentrations are denoted as $M_{IS}$, $M_{WS,}$ and $M_{BC,}$ respectively. The scalability of AeroMix in terms of the number of components that can be defined in a given mixture enabled this study to assess the possible existence of different types of core-shell mixed particles in the atmosphere. Since AeroMix models optical properties at specified RH

levels, the nearest RH value to the seasonal average of the daytime mixed layer RH for Kanpur and the regional average for BoB is chosen, respectively.

### 3.2.3. Aerosol vertical profile

The vertical distribution of aerosols in the mixed layer is modeled by fitting a cubic polynomial (Eq. (6)) to the measured $\beta_{ext}$ at 532 nm profiles. The choice to explore an alternative approach emerged because the exponential function (Eq. (5)) may not consistently capture the actual vertical variation of aerosols, particularly when there are co-existing elevated aerosol layers. This inconsistency results in an inaccurate representation of the measured $\beta_{ext}$ profiles. In the cases examined in this study, the exponential function yielded a suboptimal fit, as indicated by $R^2$ values ranging from -0.24 to 0.7 and RMSE varying between 0.03 and 0.14. Conversely, the cubic polynomial provided the best fit, with $R^2$ and RMSE values ranging from 0.91 to 0.99 and 0.004 to 0.036, respectively. The aerosol chemical composition and vertical profiles in the free troposphere and stratosphere are modeled following Hess et al. (1998). The size distributions and optical properties of all aerosol components are assumed to be uniform throughout the atmospheric column.

Assessing the impact of these assumptions about the aerosol vertical profile is challenging due to the limitation in determining the extent of changes in aerosol properties with altitude. Even if such variations are known, like other inversion algorithms (Lewandowski et al., 2010; Sinha et al., 2013), there is a limitation associated with the Mie inversion technique in estimating the effect of the changes on the inversion results, although it has been widely utilized as described above. Alternatively, sensitivity estimates on AOD of the parameters describing vertical profiles are also complicated owing to the interdependence of multiple factors influencing AOD, such as vertical profile and layer thickness of aerosols. Suppose the covariation of the aerosol vertical profile with changes in mixed layer thickness is disregarded by assuming a constant aerosol vertical profile while varying the MLH. In that case, the changes in AOD ($\Delta AOD/\Delta MLH \approx \partial AOD/\partial h$) are minimal (<0.0005) for a unit meter shift in MLH across the different vertical profiles considered (Fig. C1). A detailed description of this sensitivity analysis is given in Appendix C. However, the $\Delta AOD/\Delta MLH$ is not uniform throughout the vertical column. Instead, it follows the given vertical distribution of aerosols within the layer. This suggests that alterations in MLH have a more pronounced effect on AOD at altitudes characterized by higher aerosol concentrations than those with lower aerosol concentrations. Given the highly heterogeneous spatial and temporal nature of aerosol vertical distribution in the real atmosphere (Kumar et al., 2023), attempting a generalized quantification of the sensitivity of vertical profile assumptions on AOD lacks meaningful interpretation.

### 3.3. Assessment of probable aerosol mixing state

The probable existence of the aerosol components in the atmosphere, which refers to the aerosol mixing state is assessed by iteratively varying the number concentrations in the mixture in AeroMix until the root mean squared error (RMSE) between the measured and modeled AOD and SSA spectra are minimized. Only those spectra with RMSE minimum ≤ 0.03 are considered as the best fit (Fig. 1). The RMSE threshold of 0.03 is chosen to ensure that the RMSE remains within the lowest

uncertainty (15%) in the Aerosol Robotic Network (AERONET) AOD and SSA retrievals (Srivastava and Ramachandran, 2013). Since AeroMix models optical properties at specific wavelengths, modeled AOD values are interpolated to the measurement wavelengths for calculating the RMSE using a second-order polynomial equation (Eq. (8)) (Eck et al., 1999), which provides greater precision compared to the Ångström exponent power law.

$$ln\ AOD(\lambda) = a_2(ln\ \lambda)^2 + a_1\ ln\ \lambda + a_0 \tag{8},$$

where $\lambda$ is the wavelength at which AOD is calculated, and $a_0$, $a_1$, and $a_2$ are coefficients. However, for SSA, wavelengths closer to the measurement wavelengths are chosen in AeroMix since no equivalent relationship exists for SSA interpolation. The aerosol mixtures with modeled spectral AODs and SSAs matched well with the measured ones (RMSE≤ 0.03) are taken as probable aerosol mixing states for a given location and season under consideration. It is important to note that the aerosol mixing state modeled by the Mie inversion technique is not unique but a probable scenario. Solving a system of equations with number of unknowns greater than the constraints poses an undetermined system having multiple solutions (Sumlin et al., 2018). Hence, a physically feasible solution is selected from the AeroMix-modeled probable aerosol mixing states by further constraining with the measured component-wise $M_a$ with the modeled $M_a$ agreed to within ±1σ. Measured $M_a$ of organic carbon (OC), elemental carbon (EC), water-soluble organic carbon (WSOC), and water-soluble ionic species (WSIS) are grouped into the $M_{IS}$, $M_{WS}$, $M_{BC}$, $M_{SS,}$ and $M_{MD}$ using appropriate fixed scaling factors for each component prescribed in the literature (see Appendix B for detailed descriptions). The asymmetry parameter is not considered to constrain the mixing states; however, a brief comparison between modeled and measured asymmetry parameter values is provided in the next section.

## 4. AeroMix modeled aerosol mixing state over Kanpur and the Bay of Bengal

### 4.1. Kanpur

Aerosol mixing state over Kanpur is deduced for winter (December-February), premonsoon (March-June), and postmonsoon (October and November) seasons by constraining AeroMix modeled spectral AODs, SSAs, and $M_a$ with the measured ones obtained at Kanpur during 2007-2009. The monsoon (July-September) season is excluded from the study due to the lack of aerosol chemical composition measurements. The selection of this study period is based on the availability of collocated aerosol chemical composition data over Kanpur and the concurrent W-ICARB campaign data over the BoB during winter, of which a detailed description is provided in the following section. Due to a lack of simultaneous MPLNET observations representing the seasons under consideration, we utilized seasonal averages of MLH and $\beta_{ext}$ at 532 nm derived from MPLNET data spanning the entire operational period from May 2009 to November 2015 to model the aerosol vertical distribution.

Figures 3a-3c present the AeroMix modeled spectral AODs, SSAs, component-wise $M_a$ in external and core-shell mixed states compared with the measured ones for winter, premonsoon and postmonsoon seasons of 2007-2009 over Kanpur. The modeled percentage mass fraction of aerosols to the total $M_a$ is depicted in Fig. 3d. A summary of the probable core-shell mixed aerosol components in each season, their CSR values, and the mass fractions of components in each core-shell mixed component are

presented in Table 3. The AeroMix modeled spectral AODs and SSAs of the probable aerosol mixture agreed with the measured values for all the seasons within RMSE values of <0.03 (Figs. 3a-3b). The corresponding modeled and measured $M_a$

of all components agreed to within ±1σ (Fig. 3c). AeroMix estimated $M_{MD}$ could not be compared due to the unavailability of mineral dust measurements during the study period.

**Table 3: Probable core-shell mixed aerosol components modeled with AeroMix over Kanpur for winter, premonsoon, and postmonsoon seasons of 2007-2009 and over the BoB during winter (December 2008-January 2009). The percentage mass fraction of aerosol components participating in core-shell mixing is also presented.**

| Kanpur | | | | | | |
|---|---|---|---|---|---|---|
| Season | Core-shell mixed components | CSR | Core | | Shell | |
| | | | Component | % | Component | % |
| Winter | IS-WS | 0.8 | IS | 86.5 | WS | 49.8 |
| | BC-WS | 0.4 | BC | 51.8 | WS | 42.0 |
| | SSam-BC | 0.9 | SSam | 50.2 | BC | 9.70 |
| Premonsoon | IS-WS | 0.9 | IS | 100 | WS | 62.6 |
| | MDnm-BC | 0.3 | MDnm | 0.04 | BC | 49.6 |
| Postmonsoon | IS-WS | 0.8 | IS | 97.1 | WS | 58.6 |
| | BC-WS | 0.4 | BC | 8.0 | WS | 7.80 |
| **Bay of Bengal** | | | | | | |
| Region | Core-shell mixed components | CSR | Core | | Shell | |
| | | | Component | % | Component | % |
| W-BoB | IS-WS | 0.8 | IS | 100 | WS | 34.7 |
| | BC-WS | 0.4 | BC | 23.6 | WS | 21.9 |
| | MDnm-BC | 0.4 | MDnm | 0.70 | BC | 76.4 |
| N-BoB | IS-WS | 0.9 | IS | 100 | WS | 13.4 |
| | BC-WS | 0.4 | BC | 100 | WS | 86.6 |

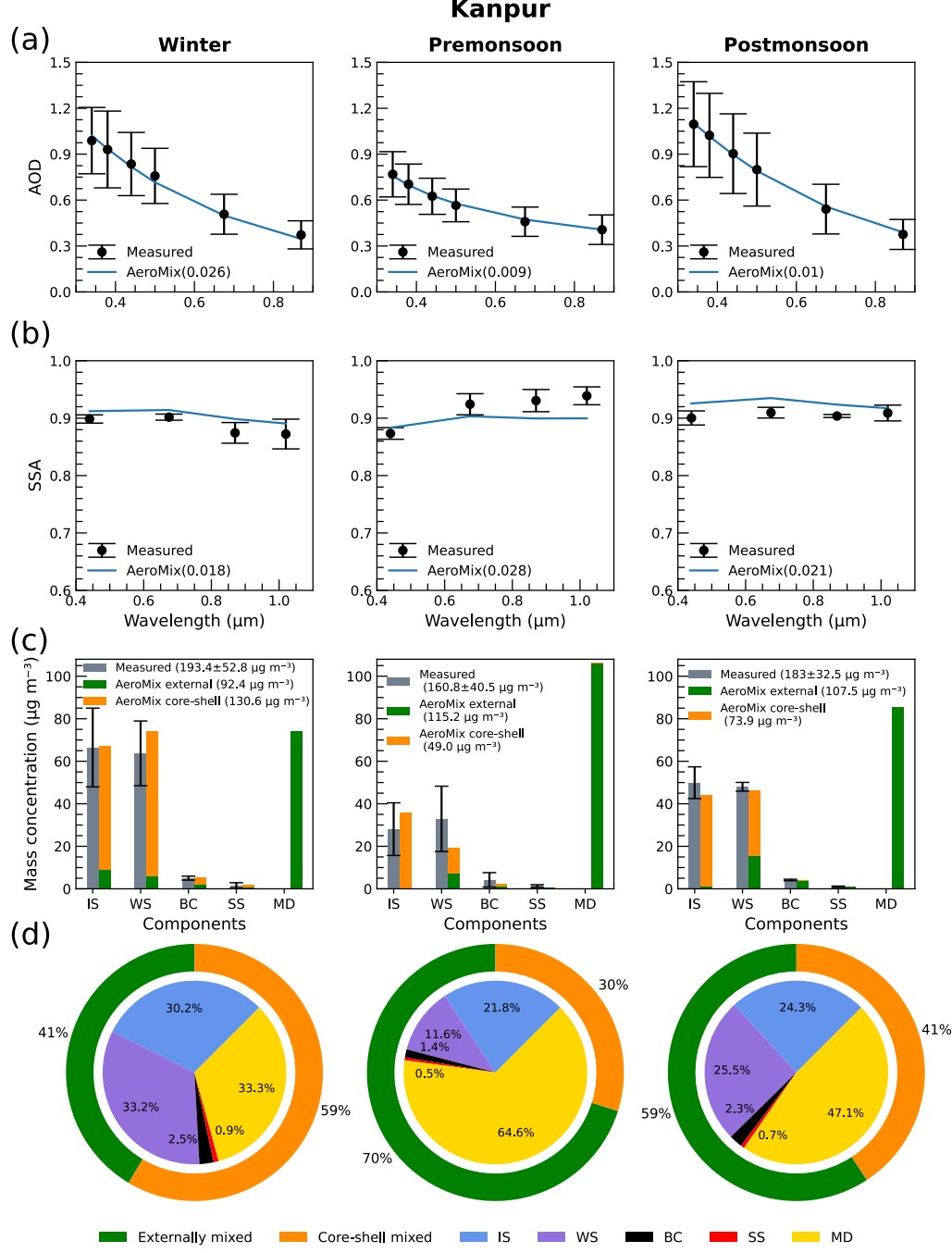

**Figure 3: Measured and AeroMix modeled aerosol parameters over Kanpur during the winter, premonsoon, and postmonsoon seasons of 2007-2009. a) AeroMix modeled season-wise spectral AOD compared with AERONET measured AOD. Vertical bars represent the standard deviation of the mean values. Root mean square error of the fit is given in parenthesis, b) Same as (a) but for SSA, c) Component-wise aerosol mass concentration in externally mixed and core-shell mixed state compared with measured aerosol mass concentration. Vertical bars represent standard deviation (±1σ) of the mean values, d) Modeled percentage mass fraction of aerosol components to the total aerosol mass (inner pie) and percentage mass fraction of externally mixed and core-shell mixed aerosols to the total aerosol mass (outer pie).**

Seasonal variability in the extent of core-shell mixing, the types of core-shell mixed particles, and the aerosol mass fraction participating in core-shell mixing are observed (Figs. 3c-d; Table 3). The percentage mass fractions of the total $M_a$ in core-shell mixed states increased from 30% in premonsoon to 59% in winter (Fig. 3d). Insoluble aerosols thinly coated with water-soluble are probable in all three seasons. Water-soluble aerosols are also found to be thickly coated over BC aerosols during the winter and postmonsoon seasons. BC aerosols are thinly coated over half of the SSam mass in winter while thickly coated over a tiny fraction of the MDnm aerosols in premonsoon season. Only SSam of sea-salt components and MDnm of mineral dust components are in the core-shell mixed state, and the majority (>99%) of mineral dust is in an externally mixed state in all seasons.

In Kanpur, the AeroMix modeled and measured spectral asymmetry parameter exhibited excellent closure for winter (RMSE = 0.024) and postmonsoon (RMSE = 0.02) but deviated slightly (RMSE = 0.065) for premonsoon (however, not shown), which could be associated with the presence of BC-coated mineral dust aerosols (Table 3) of which the non-sphericity is not accounted in AeroMix.

The earlier modeling study assessing aerosol mixing states over Kanpur also reported the probable existence of water-soluble coated BC aerosols during winter and postmonsoon, BC-coated mineral dust during premonsoon and water-soluble coated insoluble aerosols during postmonsoon seasons (Srivastava and Ramachandran, 2013). Expanding upon these findings, the present study, leveraging AeroMix capability to model any number of aerosol components in a mixture, reveals the likely coexistence of multiple core-shell mixed aerosol components in the aerosol mixture over Kanpur. The mass fractions of core and shell components involved in core-shell mixing are anticipated to differ among the studies. The present study assessed the probable coexistence of aerosol components in externally mixed states and different core-shell mixed states at various mass-mixing scenarios, considering all possibilities simultaneously. In contrast, Srivastava and Ramachandran (2013) examined the probable existence of each core-shell mixed component as a separate case, with core and shell components mixed at mass fractions varying from 20-100% of their total mass, while the rest remained externally mixed with other aerosol components. The findings in this study are further strengthened by constraining the probable mixing states with the concurrent and collocated measured aerosol chemical composition data (Ram et al., 2010a), contributing to a more robust assessment of probable aerosol mixing states.

## 4.2. Bay of Bengal

Aerosol mixing states over the W-BoB and N-BoB are analyzed for the winter season from the spectral AODs, aerosol chemical composition, MLH, and RH data measured during the W-ICARB campaign from 27[th] December 2008 to 9[th] January 2009. MLH derived from radiosonde and $\beta_{ext}$ at 532 nm from CALIOP are used to model the aerosol vertical distribution. In the absence of SSA measurement, only the measured spectral AODs and $M_a$ are used as constraints to simulate aerosol compositions and model the probable mixing state over the regions, which may influence the mixing state estimates. The modeled spectral AODs of the probable aerosol mixtures are in close agreement with the measured values (RMSE<0.03) (Fig. 4a), and modeled $M_a$ are within ±1σ of measured values for both regions (Fig. 4b).

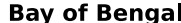

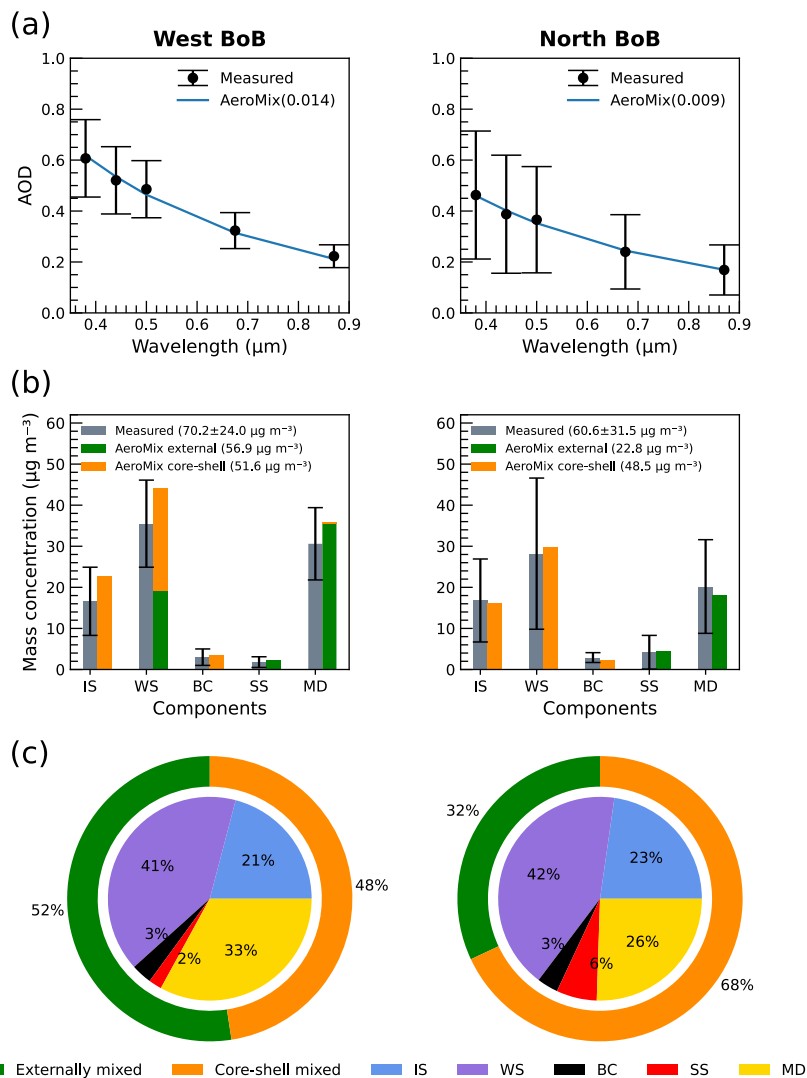

**Figure 4: Measured and AeroMix modeled aerosol parameters over the W-BoB and N-BoB during winter season (December 2008-January 2009). a) AeroMix modeled season-wise spectral AOD compared with measured AOD. Vertical bars represent the standard deviation of the mean values. Root mean square error of the fit is given in parenthesis, b) Component-wise aerosol mass concentration in externally mixed and core-shell mixed state compared with measured aerosol mass concentration. Vertical bars represent standard deviation (±1σ) of the mean values, c) Modeled percentage mass fraction of aerosol components to the total aerosol mass (inner pie) and percentage mass fraction of externally mixed and core-shell mixed aerosols to the total aerosol mass (outer pie).**

The extent of core-shell mixing is greater over the N-BoB (68%) compared to the W-BoB (48%) during the winter season. The mixtures consisting of BC-coated mineral dust; and water-soluble coated insoluble and BC particles are the probable core-shell mixing scenarios over the W-BoB. Meanwhile, over the N-BoB, only water-soluble coated insoluble and BC aerosols are probable in the core-shell mixed state (Table 3). Interestingly, over the W-BoB, about 43% of the water-soluble aerosol mass remains externally mixed, whereas, all water-soluble aerosols are coated over all of the insoluble and BC over the N-BoB. A small fraction of the mineral dust is thickly coated with BC over the W-BoB, but all of the mineral dust is externally mixed in the N-BoB. Size-segregated measurements of aerosol vertical profiles combined with airmass trajectory analysis over the BoB during W-ICARB revealed the strong presence of coarse mode aerosols in the IGP outflow (Sinha et al., 2011a). There is no similar modeling study conducted over the BoB to assess probable aerosol mixing states, making it impossible to compare the current findings directly. However, comparisons with the available aerosol mixing measurements from nearby regions at different time periods are provided in Sect. 4.4.

### 4.3. Comparison of modeled mixing states over Kanpur and Bay of Bengal during winter

The BoB, being located downwind of the IGP, is significantly affected by the continental outflow from the IGP during the winter (Fig. 2a) (Sinha et al., 2011a). During this season, a significant fraction of aerosols are in the core-shell mixed state over Kanpur and BoB. The extent of core-shell mixing is higher over the N-BoB (68%) and lower over W-BoB (48%) compared to Kanpur in winter (59%) (see Figs. 3d and 4c). The relative coating thickness of the core-shell mixed components remained similar over Kanpur and BoB, though there is a difference in the mass fraction of aerosol components participating in the core-shell mixing. The similarity in the modeled mixing states, particularly water-soluble coated insoluble and BC particles over the BoB and Kanpur suggests the transport of core-shell mixed aerosol components from the IGP to BoB during winter. Additionally, the thick coating of BC over a small fraction of mineral dust aerosols in the W-BoB region, which is not observed in Kanpur, indicates the aging effect of the long-range transport on the mixing state of the mineral dust and BC aerosols. While the $M_{WS}$ over the W-BoB is merely half of that over Kanpur, 43% of the $M_{WS}$ is in the externally mixed state over the W-BoB (Fig. 4b) compared to only 8% of $M_{WS}$ over Kanpur (Fig. 3c). This suggests a possible contribution of water-soluble aerosols (anthropogenic) to the W-BoB from proximal sources, along with continental outflow from IGP. The probable core-shell mixing scenarios presented in this study for Kanpur and the BoB are similar to those in other urban (Clarke et al., 2004; Arimoto et al., 2006) and marine regions (Guazzotti et al., 2001) and are discussed in detail in the next section.

### 4.4. Comparison of aerosol mixing states obtained with AeroMix, SP2, and Electron Microscopy

Measurements with an SP2 and an aerosol mass spectrometer (AMS) over Kanpur indicate 61.6±9.8% of the total number of BC aerosols are thickly coated in winter with low volatile oxygenated organic aerosols, ammonium, and nitrate (Shamjad et al., 2016; Thamban et al., 2017). These findings are qualitatively consistent with AeroMix, which models 51.8% of BC mass in a core-shell structure over Kanpur in winter. Due to the lack of concurrent aerosol mixing state inferred through TEM analysis at Kanpur, we qualitatively compare AeroMix and TEM-derived core-shell structures available over Delhi in the

winter of 2014 (Mishra et al., 2018). Similar aerosol characteristics over Kanpur and Delhi in the Eastern IGP are anticipated, supported by recent findings indicating dominant anthropogenic aerosols over the entire IGP region in winter (Ojha et al.,

2020). The key anthropogenic aerosols, namely BC, sulfate, and organic aerosols, generally favor the core-shell structure. In winter over Delhi, observations revealed semi-aged carbon fractal aggregates are embedded into sulfate or organic coats (Mishra et al., 2018). Other studies reported similar findings over Delhi and Varanasi in the IGP during winter (Tiwari et al., 2015; Murari et al., 2016).

Electron micrographs of the aerosol samples collected from the regions impacted by emissions from heavy industries suggest

the presence of fine metal and fly ash particles coated with secondary aerosols (Li et al., 2016), supporting the probable existence of IS-WS aerosols, as modeled by AeroMix over the heavily industrialized Kanpur throughout the year and in the outflow. Similar observations of the desert outflows at various locations have reported the presence of BC-coated MD aerosols (Arimoto et al., 2006; Clarke et al., 2004; Deboudt et al., 2010), which is probable during the premonsoon season in Kanpur and in the IGP outflow. BC aerosols coated with WS are commonly reported over urban locations (Clarke et al., 2004; Dong

et al., 2019; Li et al., 2016; Kompalli et al., 2020), agreeing with results obtained over Kanpur using AeroMix.

A collocated measurement of BC aerosols with an SP2 and aerosol chemical composition with an Aerosol Chemical Speciation Monitor (ACSM) in winter 2016-2017 over a coastal urban site of Bhubaneswar located in the IGP outflow to the W-BoB suggests coated BC aerosols (CSR ≈ 0.74), preferentially by sulfate particles (Kompalli et al., 2020). Similarly, for the continental outflow over the northern Indian Ocean from IGP via W-BoB during winter (Fig. 2a), collocated measurement

with an SP2 and ACSM reported the presence of BC aerosols thickly coated (CSR ≈ 0.37-0.59) by sulfate and organic matter (Kompalli et al., 2021). These observations are consistent with the AeroMix modeled aerosol mixing states, indicating the probable existence of the thickly coated BC with WS aerosols (CSR = 0.4) over both the W-BoB and N-BoB. This reasonable agreement between AeroMix and independent measurements, although some of these measurements are not spatially and temporally collocated, substantiates the potential capability of AeroMix to model intricate aerosol mixing states.

**5. Summary and future scope**

Atmospheric aerosols can be a complex mixture of different components due to their variety of sources, short residence time, and dynamic nature. A Python package named 'AeroMix' is developed to model the optical properties and mixing states of complex aerosol mixtures using Mie inversion. The novel features in the AeroMix encompass the capability to model both externally and core-shell (internally) mixed aerosols, simulate optical properties of aerosol mixtures comprising any number

of aerosol components, and define aerosol composition and relative humidity in up to six vertical layers. These features make AeroMix a valuable tool for modeling real atmospheric aerosol mixing scenarios by assessing the potential coexistence of aerosol components in both externally mixed states and various core-shell mixed states by considering all possibilities simultaneously.

AeroMix performance is demonstrated by modeling probable aerosol mixing states over Kanpur and the Bay of Bengal (BoB) in South Asia, representing urban and marine environments. This study presents observationally constrained mixing states of complex aerosol mixtures constituted by multiple aerosol components in externally mixed and/or core-shell mixed states with AeroMix using measured aerosol chemical and optical properties for the first time over Kanpur and BoB.

Utilizing AeroMix capability to model any number of aerosol components in a mixture, this study reveals the likely existence of multiple core-shell mixed aerosol components in the aerosol mixture over Kanpur and BoB. The AeroMix modeled aerosol mixing states are qualitatively consistent with the independent measurements using single-particle soot photometer (SP2) and transmission electron microscopy (TEM), although some of these measurements are not spatially or temporally collocated. The reasonable agreement between AeroMix and these measurements substantiates the potential capability of AeroMix to model complex aerosol mixing states involving multiple core-shell mixed components. However, this study is limited to a qualitative examination of aerosol mixing states due to the inherent constraints of the inverted Mie model approach as discussed in Sect. 3.3.

Although aerosols exhibit diverse morphology, AeroMix treats all aerosols as spherical particles except mineral dust. With the capability to incorporate various databases describing aerosol components, AeroMix ensures a better representation of aerosol properties as more refined aerosol data becomes available, accounting for morphological effects on optical properties. The prospective advancement linked to AeroMix involves the development of an optimization/machine learning algorithm utilizing AeroMix as the aerosol optical model. Such an algorithm will enable faster and more deterministic estimation of aerosol mixing states at fine temporal and spatial resolutions. Its potential applications extend beyond aerosol studies, including astrophysics and remote sensing, particularly for atmospheric correction.

**Appendix A: Evaluation of AeroMix**

The consistency of AeroMix is assessed by comparing the aerosol optical and physical properties modeled with AeroMix and OPAC (Hess et al., 1998). The component-wise mass concentration compared for ten externally mixed aerosol types given in OPAC, namely continental clean, continental average, continental polluted, urban, desert, maritime clean, maritime polluted, maritime tropical, Arctic, and Antarctic at a relative humidity (RH) of 50% is presented in Fig. A1. The AeroMix computed mass concentrations of all components are consistent with the OPAC ($r = 0.99$, slope = 1; not shown).

Similarly, the comparisons between AeroMix and OPAC modeled aerosol optical depth (AOD), single scattering albedo (SSA), and asymmetry parameter ($g$) for all wavelengths ranging from 0.25 to 40 µm are presented in Figs. A2 a-c. The optical properties such as AOD, SSA, and $g$ also agree ($r = 1$, slope = 1) with the OPAC model (Figs. A2 a-c). Since the AOD and SSA are calculated from the modeled Mie coefficients, they need not be shown separately.

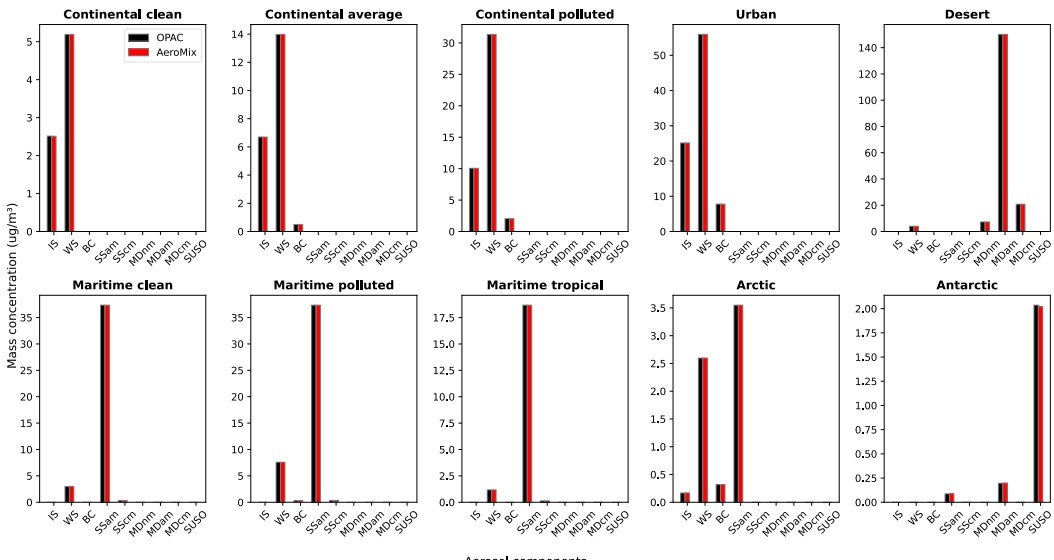

**Figure A1: Bar plots of mass concentrations of aerosol components for ten aerosol types calculated with AeroMix and OPAC at RH = 50%. The AeroMix computed mass concentrations of all components are consistent with the OPAC (r = 0.99, slope = 1; not shown).**

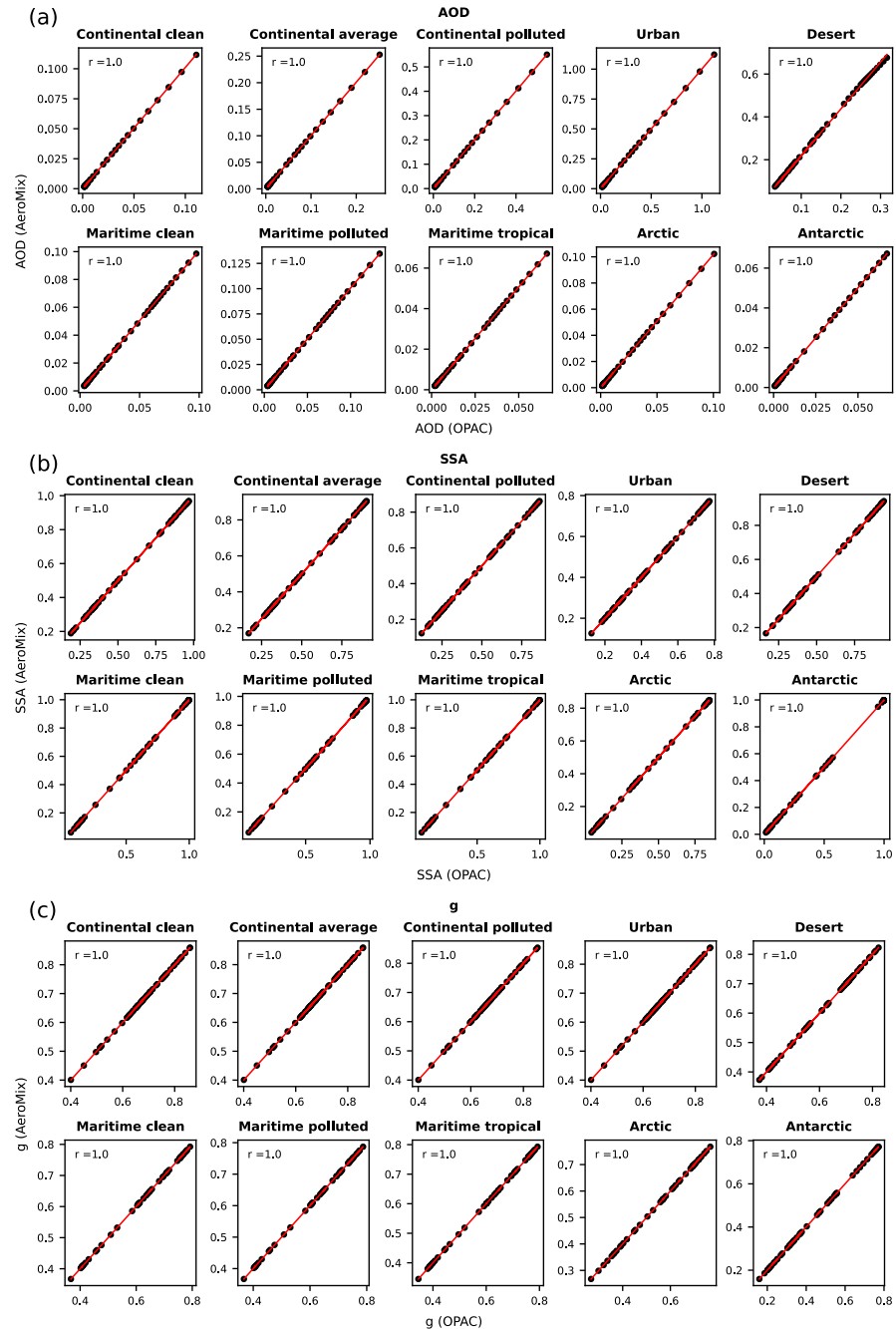

**Figure A2: Scatter plots of AeroMix calculated values of a) aerosol optical depth (AOD), b) single scattering albedo (SSA) and c) asymmetry parameter (g) for ten aerosol types at RH = 50% for all wavelengths with those calculated by OPAC. The solid red line is the least-squares regression line forced through the origin (intercept = 0).**

## Appendix B: Study region and data

### B.1. Kanpur

Kanpur (26.513° N, 80.232° E, 123 m AMSL), located at the central part of the Indo-Gangetic Plain (IGP), experiences
moderate to high levels of pollution resulting from both anthropogenic and natural emissions (Ram et al., 2010a).

To assess the probable mixing state and chemical composition of aerosols, monthly level 2 (cloud screened and quality assured) spectral AOD (0.34, 0.38, 0.44, 0.5, 0.675, and 0.87 µm) and spectral SSA (0.44, 0.675, 0.87, and 1.02 µm) measurements obtained from the AERONET (Holben et al., 2001) station at Kanpur during the period January 2007 to December 2009 are utilized. The total uncertainty associated with the measured AOD values under cloud-free conditions is estimated to be $<\pm0.01$
for wavelengths $\geq440$ nm and $<\pm0.01$ for shorter wavelengths (Smirnov et al., 2000). AOD measured at 1.02 µm is not considered for the analysis to avoid the effect of water vapor absorption.

Daytime mixed layer height (MLH) and aerosol extinction coefficient ($\beta_{ext}$) data are obtained from the NASA Micro-Pulse Lidar Network (MPLNET) Planetary Boundary Layer and Aerosol products (version 3, level 1.5), respectively (Welton et al., 2001; Lewis et al., 2013) to model the vertical distribution of mixed layer aerosols in AeroMix. The MPLNET data available
over Kanpur only covers five months (May-June and October-December 2009) during the study period. Due to the lack of MPLNET data over Kanpur representing all months during the study period, seasonal averages of MLH and $\beta_{ext}$ at 532 nm derived during the entire operational period of MPLNET, spanning from May 2009 to November 2015, are utilized.

The measured mass concentration of aerosols over the Kanpur is obtained from the $PM_{10}$ aerosol samples collected during daytime (~8-10 hours) using a high-volume sampler operated at a flow rate of $1.0 \pm 0.1$ m$^3$min$^{-1}$ on pre-combusted tissuquartz
(PALLFLEX™) filters of size $20.0 \times 25.4$ cm$^2$ (Ram et al., 2010b). Sixty-six samples were collected during the period January 2007 to March 2008 and were analyzed for organic carbon (OC), elemental carbon (EC), water-soluble organic carbon (WSOC), and water-soluble ionic species (WSIS). No samples were collected during the monsoon season (July-September). A detailed description of the analytical procedures and the data can be found in Ram et al. (2010a). The measured PM10 aerosol chemical composition is grouped into water-insoluble (IS), water-soluble (WS), black carbon (BC), sea-salt (SS), and
mineral dust (MD). The mineral dust mass concentration is calculated by scaling the measured Aluminum concentration by a factor of 12.5 since the soil mass contains 8.04% of Aluminum (Srinivas et al., 2011; Taylor and McLennan, 1985). However, mineral dust mass concentration is not calculated and validated for Kanpur due to the unavailability of measured Al mass concentration. The sea-salt mass concentration is estimated by scaling the measured Na$^+$ mass concentration with a factor of 3.2, considering the salinity of seawater to be ~35g kg$^{-1}$ (Millero et al., 2008). Measured EC is taken as the BC concentration.
WSIS and WSOC constitute the WS mass concentration, and OC×1.6 is taken as the IS mass concentration.

Daytime relative humidity (RH) values are obtained from the automatic weather station installed at Kanpur (www.mosdac.gov.in/catalog/insitu.php) for the period November 2007 to December 2009. The availability of the collocated aerosol chemical, optical, and vertical profile measurements make Kanpur desirable for assessing the aerosol mixing states representing an urban location.

## B.2. Bay of Bengal

BoB is located in the northeastern part of the Indian Ocean, bounded by the Indian subcontinent to the west and the Indochinese Peninsula to the east. Anthropogenic and natural emissions from these surrounding landmasses largely influence the aerosol characteristics over the BoB during winter seasons (Srinivas et al., 2011). The northwestern part of the BoB, which is most affected by the continental outflow of aerosols during the winter (Sinha et al., 2011a), is selected to assess the aerosol mixing state over the marine environment.

The probable aerosol mixing state over the western and northern part of the BoB (W-BoB and N-BoB) during the winter season are derived from the data obtained from the Winter-Integrated Campaign for Aerosols, gases and Radiation Budget (W-ICARB) conducted from 27th December 2008 to 30th January 2009 (Moorthy et al., 2010). The W-BoB and N-BoB are demarcated following Kaskaoutis et al. (2011). The spectral AOD, MLH, RH, and aerosol chemical composition data measured from 27th December 2008 to 9th January 2009 are used in this study.

Spectral AOD (0.38, 0.44, 0.5, 0.675, and 0.87 µm) measurements were made using a handheld sunphotometer MICROTOPS-II (Solar Light Company, USA) at ~10-minute intervals. Th e estimated uncertainty in AOD measurements made by the MICROTOPS-II is less than ±0.03 (Morys et al., 2001; Smirnov et al., 2009). Only cloud-free data are used for the analysis, and triplet observations were made to avoid any possible operator error in sun pointing on the moving platform. For a detailed description of AOD measurements and quality controls, refer to Kaskaoutis et al. (2011).

MLH values are derived from thermodynamics variables, namely, potential temperature, wind shear, specific humidity, and bulk Richardson number. These parameters are derived from Pisharoty GPS radiosonde (make: VSSC, ISRO, India) launched daily three times at 00:30, 06:30, and 13:30 local time (LT) during the campaign. Daily mean MLH derived from 22 morning (06:30 LT) and afternoon (13:30 LT) launches were utilized in this study. A detailed description of radiosonde and methodology of the derivation of MLH are presented in Subrahamanyam et al. (2012) and thus not repeated here.

Cloud-Aerosol Lidar with Orthogonal Polarization (CALIOP) level 3, V4 cloud-free daytime extinction coefficients at 532 nm (Kim et al., 2018) were used to model the tropospheric aerosol profile over the BoB. CALIOP Level 3 extinction coefficient at 532 nm is a monthly averaged globally gridded data based on quality-screened level 2 aerosol extinction profiles up to an altitude of 12 km. Cloud-free extinction coefficient profiles are quality screened following Tackett et al. (2018) and aggregated uniformly gridded onto a global 2° latitudes × 5° longitudes with a vertical resolution of 60 m. The incorrect LiDAR ratios (Omar et al., 2009) and aerosol type classification (Huang et al., 2013) are the two important contributors to the uncertainty in CALIOP Level 3 aerosol extinction. Despite these uncertainties, Winker et al. (2013) have shown that these Level 3 aerosol data are realistic and representative of aerosol extinction greater than about 0.001 $km^{-1}$ and up to an altitude of 6 km. Thus, we excluded the extinction coefficient values below 0.001 $km^{-1}$ in our analyses. In addition, the CALIOP Level 3 extinction coefficient has been extensively used in investigating the seasonal evolution of extinction profile (Huang et al., 2013), global aerosol source attribution (Prijith et al., 2016), estimates of wildfire injection heights (Sofiev et al., 2013), and aerosol radiative effect (Chung et al., 2016).

Air samples containing $PM_{10}$ aerosols were collected during the W-ICARB campaign over the BoB onboard the ship using a high-volume sampler (Thermo Andersen, USA) 15 m above the mean sea level at a flow rate of about one $m^3min^{-1}$ with a variation of 5%. Each sample was collected over a time period ranging from 20 to 22 h on pre-combusted tissuquartz (PALLFLEX$^{TM}$) filters of size $20\times25$ cm$^2$. A total of 11 samples were collected during the cruise over the W-BoB and N-BoB and were analyzed for OC, EC, WSOC, WSIS, trace metals (Cd and Pb), and crustal constituents (Al, Fe, Ca, and Mg). Detailed descriptions of the analytical procedures are given elsewhere (Srinivas et al., 2011).

RH measurements taken during the cruise using meteorological sensors onboard the ship are used to model the aerosol hygroscopic growth.

## Appendix C: Sensitivity of AOD to layer thickness

To evaluate the impact of changes in layer thickness on AOD, sensitivity analyses are conducted, keeping the vertical profiles of aerosols constant while varying MLH. Figure C1 presents the variations in mixed layer AOD at 0.5 µm in response to alterations in MLH ($\Delta AOD/\Delta MLH$). The AOD is calculated for the urban aerosol type (IS (number concentration) = 1.5 cm$^{-3}$, WS = 28000 cm$^{-3}$, and BC = 130000 cm$^{-3}$) at a relative humidity of 50% using AeroMix.

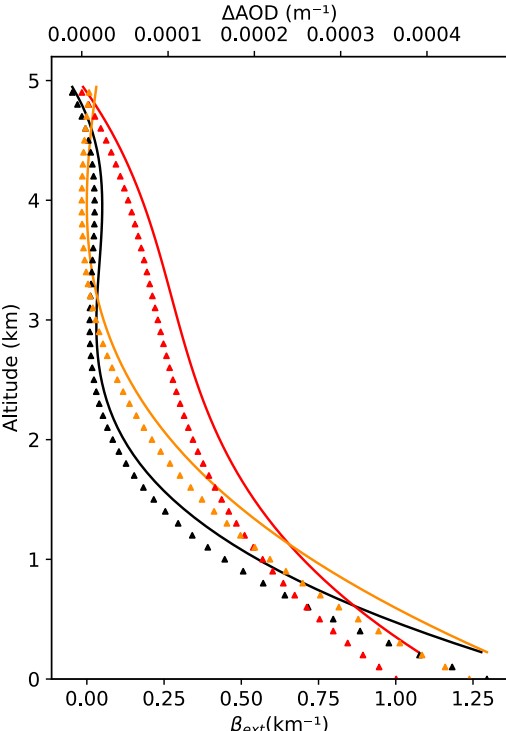

**Figure C1: Variation in mixed layer aerosol optical depth (AOD) per unit meter change in mixed layer height (MLH) across various sample vertical profiles. Calculations were performed using AeroMix for an urban aerosol type. Solid lines illustrate the sample vertical profiles, while the triangles indicate the corresponding AOD changes per unit meter change in MLH.**

## Code availability

The AeroMix package is publicly available on Zenodo (https://doi.org/10.5281/zenodo.10552079) and GitHub
(https://github.com/sampr7/AeroMix) under GNU General Public License-3.0 (P Raj and Sinha, 2024a). The key package
dependencies include,

1. NumPy (www.numpy.org)
2. SciPy (ww.scipy.org)
3. PyMieScatt (www.github.com/bsumlin/PyMieScatt).

The model outputs and codes used to generate the results presented are available in a separate Zenodo repository:
https://doi.org/10.5281/zenodo.12625421 (P Raj and Sinha, 2024b).

## Data availability

The AERONET level 2 data (www.aeronet.gsfc.nasa.gov/), MPLNET data (www.mplnet.gsfc.nasa.gov/), CALIOP data
(www.subset.larc.nasa.gov/calipso/), RH data over Kanpur (www.mosdac.gov.in/catalog/insitu.php), and 10 m surface wind
data (https://cds.climate.copernicus.eu/cdsapp#!/dataset/reanalysis-era5-single-levels-monthly-means?tab=overview) are
openly available from respective websites. The aerosol chemical composition over Kanpur (Ram et al., 2010a), AOD
(Kaskaoutis et al., 2011), aerosol chemical composition (Srinivas et al., 2011), and mixed layer height (Subrahamanyam et al.,
2012) data collected during W-ICARB campaign are presented in respective sources.

## Author Contribution

PRS conceived the ideas and designed the study. SPR developed the AeroMix package, performed analysis, and prepared the
original draft with inputs from PRS. RS contributed to the scientific discussions on mixing state calculations. DBS and SB
provided the radiosonde and chemical composition data, respectively, obtained during the W-ICARB cruise and contributed
to the scientific discussions.

## Competing interests

The authors declare that they have no conflict of interest.

## Acknowledgements

This work was supported as part of developing the IIST Ponmudi Climate Observatory under ASRG-ISRO program. We
acknowledge the PIs of Kanpur AERONET and MPLNET for their effort in establishing and maintaining the sites. The authors

gratefully acknowledge the ISRO-GBP program office and SPL-VSSC for conducting the W-ICARB campaign and are
thankful to IMD for the RH data collected during the campaign. SPR sincerely acknowledges B L Madhavan for his continuous guidance and insightful discussions. The authors thank Simon O'Meara and the two other anonymous reviewers for their constructive comments which helped to improve the overall presentation of this paper.

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
