# Peer review of "AeroMix v1.0.1: a Python package for modeling aerosol optical properties and mixing states"

_EGUsphere, 2024_

## Author Comment (AC1)

**Response notes**

**Manuscript Title**: AeroMix v1.0.1: a Python package for modeling aerosol optical properties and mixing states [egusphere-2024-62]

**Reviewer: # 1**

**General comments**

The study fits into the scope of GMD. Minor organization of the methods section is needed as described below.

**Reply:** We appreciate and thank the reviewer's valuable and detailed comments and recommendation. All comments have been taken into account, as described below. We have incorporated the necessary corrections in the revised manuscript being submitted.

**Introduction**

1. When discussing motivation, the mixing state of aged aerosol (far from source) is important for CCN (Farmer, Cappa, and Kreidenweis 2015). McFiggans et al. 2006 is another source discussing CCN from close to source and aged populations.

   **Reply:** As per your suggestion the text is revised as;

   *"Knowledge of the size-resolved aerosol chemical composition, size distribution, and mixing state is required to predict cloud condensation nuclei (CCN) concentrations for freshly emitted and aged aerosols (McFiggans et al., 2006; Ervens et al., 2010; Farmer et al., 2015)."*

**Model Overview**

2. For the readers' convenience, add a table for chemical species modeled (IS, WS, BC, SS, MD). Table columns should include the name, shorthand name (e.g. IS), species considered/represented (IS includes soil dust, fly ash, and non-hygroscopic organic matter from biomass burning), and modes present (nucleation (nm), accumulation (am), coarse (cm)).

   **Reply:** A new table presenting the compositions of predefined components, their shorthand name, constituting species, along with their dry state (0% RH) size distribution parameters and refractive index (at 0.5 µm) is added. The discussion about constituting species in the lines from #83-91 is removed to avoid redundancy.

**Table 1: Size distribution parameters, specific density and refractive index of predefined aerosol components at dry state (0% RH).**

| Aerosol component | Constituting species | Mode radius (µm) | Standard deviation | specific density (g cm$^{-3}$) | Refractive index at 0.5 µm |
|---|---|---|---|---|---|
| Insoluble (IS) | Soil dust, fly ash, and non-hygroscopic organic matter. | 0.471 | 2.51 | 2.0 | $1.53 + 8 \times 10^{-3}$i |
| Water-soluble (WS) | $SO_4^{2-}, NO_3^-, NH_4^+$ and | 0.0212 | 2.24 | 1.8 | $1.53 + 5 \times 10^{-3}$i |

| | | | | | |
|---|---|---|---|---|---|
| | hygroscopic fraction of organic matter. | | | | |
| Black carbon (BC) | Black carbon aerosols | 0.0118 | 2 | 1.8 | $1.75 + 4.5 \times 10^{-1}i$ |
| Sea-salt accumulation mode (SSam) | Sea salt aerosols | 0.209 | 2.03 | 2.2 | $1.5 + 1.55 \times 10^{-8}i$ |
| Sea-salt coarse mode (SSam) | | 1.75 | 2.03 | | |
| Mineral dust nucleation mode (MDnm) | Desert dust | 0.007 | 1.95 | 2.6 | $1.53 + 7.8 \times 10^{-3}i$ |
| Mineral dust accumulation mode (MDam) | | 0.39 | 2 | | |
| Mineral dust coarse mode (MDcm) | | 1.9 | 2.15 | | |
| Stratospheric sulfate (SU) | Sulfate aerosols from volcanic eruption. | 0.0695 | 2.03 | 1.7 | $1.431 + 1 \times 10^{-8}i$ |

3. It is unclear if the optical properties for coated MD particles are calculated using Mie theory or TMM. Clarify if TMM only applies to particles comprised of 100% MD.

**Reply:** Optical properties of particles composed solely of MD are calculated using the TMM code. These values are adopted from Koepke et al. (2015). To avoid further any confusion, the component names is mentioned explicitly. These texts are revised as

*"The optical properties of each component, except for MDnm, MDam, and MDcm components, are calculated using the Mie theory (Mie, 1908). This calculation assumes that the particles are spherical in shape and follow a lognormal size distribution. The calculated optical properties are normalized to one particle cm$^{-3}$ and stored, which can then be scaled to any given number concentration. MDnm, MDam, and MDcm components are modeled using the T-Matrix Method (TMM) (Waterman, 1971; Mishchenko et al., 1999) to account for their non-sphericity (Koepke et al., 2015)."*

Optical properties of all the core-shell mixed particles are modeled using the coated-sphere Python Mie calculation program PyMieScatt; which is mentioned in the revised manuscript (lines # 116-119).

4. The exponential function for the vertical profile of aerosol concentration is mentioned in Eq. 5; however, only the cubic function is utilized in the modeling section considering it provides the better fit based on R2 and RMSE values. The authors do not necessarily need to include the exponential function since it is not used to model vertical profiles in this paper.

**Reply:** We prefer to retain the exponential function to model aerosol vertical profile. It is primarily due to the considerations aimed at offering flexibility in the AeroMix model to adopt under different scenarios based on the following considerations:

a. The predecessor model, OPAC, uses the exponential function to model the aerosol vertical profile. Retaining this function allows for comparisons between AeroMix-modeled AOD values with OPAC-modeled ones, facilitating consistency in analyses.

b. The cubic function is employed for modeling due to its superior fit based on $R^2$ and RMSE values described in this paper. However, it may not always be applicable in locations lacking aerosol vertical profile measurements. Therefore, selecting suitable generalized exponential functions for the aerosol mixture (e.g., marine, urban, desert, continental) becomes necessary.

c. Offering users the option to choose from the given three types of aerosol vertical profiles (exponential, cubic, and homogeneous) enhances flexibility, enabling selection based on the specific study location and layer characteristics.

The section is revised as following to provide more clarity;

"*The vertical profile of aerosol concentration in each layer can be modeled as homogenous or as an exponential function given by,*

$$N(h) = N(0)exp\left(-h/z\right), \tag{5}$$

*where N(0) is the number concentration of aerosol at the layer bottom, h is the height from the layer bottom in kilometers, and z is the scale height in kilometers representing the change in aerosol concentration with height (Hess et al., 1998). Standard exponential profiles for different aerosol types provided by Hess et al. (1998) can be utilized for locations lacking aerosol vertical profile measurements. Alternatively, when measured aerosol profiles are available, they can be modeled using a cubic function given by,*

$$N(h) = N(0)(ah^3 + bh^2 + ch + d), \tag{6}$$

*where a, b, c, and d are the coefficients for $N(0) = 1$ (Russo et al., 2006)."*

We hope you will agree that maintaining the exponential function alongside other options to provide users a comprehensive package, ensuring adaptability to diverse research contexts.

5. This section was jumpy and difficult to follow. I suggest further dividing into subsections:

**Reply:** In accordance with your suggestion, we have carefully divided it into subsections as follows:

"

[revised manuscript text omitted]

,,

6. Little mention of future work/applications. Maybe mention potential model development projects and associated improvement to probable mixing state predictions.

**Reply:** The lines #404-408 is revised as;

*"The prospective advancement linked to AeroMix involves the development of an optimization/machine learning algorithm utilizing AeroMix as the aerosol optical model. Such an algorithm will enable faster and more deterministic estimation of aerosol mixing states at fine temporal and spatial resolutions."*

**Minor comments**

**Line 43:** Healy et al. 2014 and Ye et al. 2018 describe in situ measurements of mixing state between external and internal.

**Reply:** The single-particle mass spectrometry observations presented in these studies will help present the case in our manuscript. Suggested references are incorporated to strengthen the discussion on real atmospheric aerosol mixing states. The sentence is revised as;

*"Despite the experimental evidences that the aerosol mixing state lies between a purely externally mixed state and a purely internally mixed state (Healy et al., 2014; Li et al., 2016; Ye et al., 2018; Riemer et al., 2019), the standard FORTRAN-based OPAC model considers the external mixing of aerosols alone and cannot treat complex aerosol internal mixing states."*

**Line 58:** Unclear to reader why "batch mode" would have limited functionalities. Either clarify or reword.

**Reply:** We removed this discussion about AeroGUI to avoid ambiguity.

**Line 97-98:** Unclear why density of BC is mentioned here.

**Reply:** The specific densities for all components are sourced from the Global Aerosol Data Set and OPAC database. However, the observational studies on black carbon suggest using a specific density value of 1.8 g cm$^{-3}$, differing from the 1.0 g cm$^{-3}$ value provided in OPAC. This clarification will be added to the manuscript as follows**:**

*"The density of BC is set at 1.8 g cm$^{-3}$ supported by observations, diverging from the 1.0 g cm$^{-3}$ given in OPAC  (Bond et al., 2013; Kondo, 2015)."*

**Line 137 – 139:** This sentence is confusing, specifically "and particles composed of multiple chemical species (internally mixed) as an external mixture."

**Reply:** The sentence is revised as follows for brevity.

*"Optical properties of the complex aerosol mixing states are modeled by accounting for any number of both externally mixed particles (composed of single chemical species) and internally mixed*

*particles (composed of multiple chemical species), with no presumed chemical or physical interaction among the particles within the mixture (see Fig. 1)."*

**Equation (7):** I believe index of summation should be layer, not n.

**Reply:** The equation is corrected as,

"

$$Total\ AOD = \sum_{n=1}^{6} \beta_{ext_n} \int_{h_{min_n}}^{h_{max_n}} N_n(h)dh$$

where $h_{min}$ and $h_{max}$ are the layer bottom and layer top height for each layer $n$."

**Line 163:** Consider adding one sentence on the Mie inversion technique.

**Reply:** The sentence is revised as

*"The probable aerosol mixing states are modeled with AeroMix using the widely used Mie inversion technique by iteratively comparing the multispectral measurements of aerosol optical properties with modeled ones for different mixing scenarios until they converge within the observational error (Fig. 1) (Chandra et al., 2004; Dey et al., 2008; Kaskaoutis et al., 2011; Ramachandran and Srivastava, 2016; Srivastava et al., 2016, Srivastava et al., 2018)"*

**Figure 2:** Consider adding average wind directions for winter, pre-monsoon and post-monsoon. This would aid in the comparison of modeled mixing states discussion (Sections 4.3 and 4.4). Label IGP region.

**Reply:** Average wind directions for different seasons are added to Figure 2, along with labeling the IGP region as shown below.

[Figure]

**Figure 2. a) Location of Kanpur (diamond) and the W-ICARB cruise track (dotted lines). Measurements of AOD, chemical composition, and relative humidity over the West-BoB and North-BoB (demarcated with dashed lines) were utilized in this study. Additionally, the locations of Delhi, Varanasi, and Bhubaneswar (circles) are indicated, where mixing state measurements were obtained for comparison with the modeled mixing states. The shaded area marks the Indo-Gangetic Plain. Seasonal average surface winds over the region from ERA5 reanalysis are shown for a) winter, b) premonsoon and c) postmonsoon by vectors.**

**Line 243:** Add reasoning for RMSE threshold after sentence: "Only those spectra with RMSE minimum 0.03 are considered as the best fit (Fig. 1). [Add here]." The last sentence of this paragraph "The RMSE threshold of 0.03 is chosen to ensure …" should be moved to this location.

**Reply:** Revised as suggested.

**Figure 3 and 4:** Unnecessary to include colors for externally mixed and core-shell mixed in caption.

**Reply:** Revised as suggested.

**Technical corrections**

**Line 11:** Add comma after "aerosol chemical compositions,"

**Reply:** Corrected as suggested.

**Line 13:** Remove "the" so the sentence reads "While limitations in the measurements of aerosol …"

**Reply:** The sentence is modified as per the following comment.

**Line 13-15:** Sentence "While the limitations in the measurements …" is a run on. Consider breaking into two sentences.

**Reply:** Following your suggestion, we corrected as the sentence as:

*"Limitations in the direct measurements of aerosol chemical composition and mixing states necessitate modeling approaches to infer the aerosol mixing states. The Optical Properties of Aerosols and Clouds (OPAC) model has been widely used to construct optically equivalent aerosol chemical compositions from measured aerosol optical properties using Mie inversion."*

**Line 18:** Move position of "3)" – "components, and 3) define aerosol composition"

**Reply:** Corrected as suggested.

**Line 98:** Add comma "The optical properties of each component, except for the MD components, are calculated"

**Reply:** Corrected as suggested.

**Line 209:** Run-on sentence, rewrite as "…CSR values of core-shell mixed aerosols in the model. This offers a more flexible approach compared to relying on measured…"

**Reply:** Corrected as suggested.

**Line 297:** Change to "… mixed states varied from 30% to 59% in premonsoon and winter, **respectively."**

**Reply:** Changing to "mixed states varied from 30% to 59% in premonsoon and winter, respectively" may lead to confusion regarding whether these values represent intra-seasonal or inter-seasonal variation. To clarify, we propose revising it to:

"*The percentage mass fractions of the total $M_a$ in core-shell mixed states increased from 30% in premonsoon to 59% in winter (Fig. 3d).*"

---

## Author Comment (AC2)

**Response notes**

**Manuscript Title**:  AeroMix v1.0.1: a Python package for modeling aerosol optical properties and mixing states [egusphere-2024-62]

**Reviewer: # 2**

**This manuscript is generally well-composed.**

**Reply:** We appreciate and thank the reviewer for his/her constructive comment and recommendation. We have incorporated the necessary corrections in the revised manuscript being submitted.

**Dividing the content between lines 80 to 120 into more succinct paragraphs could significantly improve readability.**

**Reply:** Following your suggestion, we have divided section 2 into two subsections and added a table to explain pre-defined components to improve readability. The section 2 is restructured and revised as follows.

"

[revised manuscript text omitted]

$$N(h) = N(0)(ah^3 + bh^2 + ch + d), \tag{6}$$

where $a$, $b$, $c$, and $d$ are the coefficients when $N(0) = 1$ (Russo et al., 2006). The default values of aerosol concentration, RH, and profile type for the free troposphere, stratosphere, and elevated mineral dust layer are adopted from Hess et al. (1998). The total column AOD is calculated by,

$$Total\ AOD = \sum_{n=1}^{6} \beta_{ext_n} \int_{h_{min_n}}^{h_{max_n}} N_n(h)dh$$

$$\tag{7}$$

where $h_{min}$ and $h_{max}$ are the layer bottom and layer top height for each layer $n$.

The AeroMix package and detailed documentation are available online at www.github.com/sampr7/AeroMix (P Raj and Sinha, 2024a).

,,

---

## Author Response (AR1)

**Cover letter**

To

Dr. Sylwester Arabas
Topic Editor,
Geoscientific Model Development (GMD)

15 May 2024

**Subject: Final author reply to the editor.**
Manuscript Title:  AeroMix v1.0.1: a Python package for modeling aerosol optical properties and mixing states.
Manuscript ID: EGUSPHERE-2024-62

Dear Dr. Sylwester Arabas,

Along with this letter, we submit the revised version of our manuscript titled "AeroMix v1.0.1: A Python package for modeling aerosol optical properties and mixing states" to the Journal of Geoscientific Model Development (GMD).

The co-authors and I appreciate the constructive comments on this manuscript by the reviewers. The comments have been very thorough and valuable in improving the manuscript. We have taken them fully into account while revising the manuscript. Our detailed responses to the comments are attached, with line numbers referenced as they appear in the tracked changed version.

All the authors agreed with the revision of this manuscript.

I request you to kindly take the necessary steps for the manuscript's earliest review.

Thank you.

Sincerely yours,

P. R. Sinha
Department of Earth and Space Sciences
Indian Institute of Space Science and Technology (IIST)
Valiamala, P.O., Thiruvananthapuram-695 547, India
Phone No. (Off): +91-471-2568523
e-mail id: prs@iist.ac.in

**Response notes**

**General comments**

The study fits into the scope of GMD. Minor organization of the methods section is needed as described below.

**Reply:** We appreciate and thank the reviewer's valuable and detailed comments and recommendation. All comments have been taken into account, as described below.

**Introduction**

1. When discussing motivation, the mixing state of aged aerosol (far from source) is important for CCN (Farmer, Cappa, and Kreidenweis 2015). McFiggans et al. 2006 is another source discussing CCN from close to source and aged populations.

   **Reply:** As per your suggestion the text is revised as (Page #2, Line #33).

   *"Knowledge of the size-resolved aerosol chemical composition, size distribution, and mixing state is required to predict cloud condensation nuclei (CCN) concentrations for freshly emitted and aged aerosols (McFiggans et al., 2006; Ervens et al., 2010; Farmer et al., 2015)."*

**Model Overview**

2. For the readers' convenience, add a table for chemical species modeled (IS, WS, BC, SS, MD). Table columns should include the name, shorthand name (e.g. IS), species considered/represented (IS includes soil dust, fly ash, and non-hygroscopic organic matter from biomass burning), and modes present (nucleation (nm), accumulation (am), coarse (cm)).

   **Reply:** A new table presenting the compositions of predefined components, their shorthand name, constituting species, along with their dry state (0% RH) size distribution parameters and refractive index (at 0.5 µm) is added (Page #7). The discussion about constituting species in the lines from #99-107 is removed to avoid redundancy.

**Table 1: Size distribution parameters, specific densities, and refractive indices of predefined aerosol components at dry state (0% RH).**

| Aerosol component | Constituting species | Mode radius (µm) | Standard deviation | specific density (g cm$^{-3}$) | Refractive index at 0.5 µm |
|---|---|---|---|---|---|
| Insoluble (IS) | Soil dust, fly ash, and non-hygroscopic organic | 0.471 | 2.51 | 2.0 | $1.53 + 8 \times 10^{-3}i$ |

| | | | | | |
|---|---|---|---|---|---|
| | matter. | | | | |
| Water-soluble (WS) | $SO_4^{2-}$ , $NO_3^-$ , $NH_4^+$ and hygroscopic fraction of organic matter. | 0.0212 | 2.24 | 1.8 | $1.53 + 5 \times 10^{-3}$i |
| Black carbon (BC) | Black carbon aerosols | 0.0118 | 2 | 1.8 | $1.75 + 4.5 \times 10^{-1}$i |
| Sea-salt accumulation mode (SSam) | Sea salt aerosols | 0.209 | 2.03 | 2.2 | $1.5 + 1.55 \times 10^{-8}$i |
| Sea-salt coarse mode (SSam) | | 1.75 | 2.03 | | |
| Mineral dust nucleation mode (MDnm) | Desert dust | 0.007 | 1.95 | 2.6 | $1.53 + 7.8 \times 10^{-3}$i |
| Mineral dust accumulation mode (MDam) | | 0.39 | 2 | | |
| Mineral dust coarse mode (MDcm) | | 1.9 | 2.15 | | |
| Stratospheric sulfate (SU) | Sulfate aerosols from volcanic eruption. | 0.0695 | 2.03 | 1.7 | $1.431 + 1 \times 10^{-8}$i |

3. It is unclear if the optical properties for coated MD particles are calculated using Mie theory or TMM. Clarify if TMM only applies to particles comprised of 100% MD.

**Reply:** Optical properties of particles composed solely of MD are calculated using the TMM code. These values are adopted from Koepke et al. (2015). To avoid further confusion, the component names are mentioned explicitly. The lines #119-124 (Pages #5&6) are revised as

"The optical properties of each component, except for MDnm, MDam, and MDcm components, are calculated using the Mie theory (Mie, 1908). This calculation assumes that the particles are spherical in shape and follow a lognormal size distribution. The calculated optical properties are normalized to one particle cm$^{-3}$ and stored, which can then be scaled to any given number concentration. MDnm, MDam, and MDcm components are modeled using the T-Matrix Method (TMM) (Waterman, 1971; Mishchenko et al., 1999) to account for their non-sphericity (Koepke et al., 2015)."

Optical properties of all the core-shell mixed particles are modeled using the coated-sphere Python Mie calculation program PyMieScatt; as indicated in the manuscript (Page #8, Line #149).

4. The exponential function for the vertical profile of aerosol concentration is mentioned in Eq. 5; however, only the cubic function is utilized in the modeling section considering it provides the better fit based on R2 and RMSE values. The authors do not necessarily need to include the exponential function since it is not used to

model vertical profiles in this paper.

**Reply:** We prefer to retain the exponential function to model the aerosol vertical profile. It is primarily due to the considerations aimed at offering flexibility in the AeroMix model to adopt under different scenarios based on the following considerations:

    a. The predecessor model, OPAC, uses the exponential function to model the aerosol vertical profile. Retaining this function allows for comparisons between AeroMix-modeled AOD values with OPAC-modeled ones, facilitating consistency in analyses.

    b. The cubic function is employed for modeling due to its superior fit based on $R^2$ and RMSE values described in this paper. However, it may not always be applicable in locations lacking aerosol vertical profile measurements. Therefore, selecting suitable generalized exponential functions for the aerosol mixture (e.g., marine, urban, desert, continental) becomes necessary.

    c. Offering users the option to choose from the given three types of aerosol vertical profiles (exponential, cubic, and homogeneous) enhances flexibility, enabling selection based on the specific study location and layer characteristics.

The lines #179-188 (Page #9) are revised as following to provide more clarity.

"The vertical profile of aerosol concentration in each layer can be modeled as homogenous or as an exponential function given by,

$$N(h) = N(0)exp\left(-h/z\right), \qquad (1)$$

where $N(0)$ is the number concentration of aerosol at the layer bottom, $h$ is the height from the layer bottom, and $z$ is the scale height representing the change in aerosol concentration with height (Hess et al., 1998). Standard exponential profiles for different aerosol types provided by Hess et al. (1998) can be utilized for locations lacking aerosol vertical profile measurements. Alternatively, it can be modeled using a cubic function (Eq. (6)) when aerosol vertical profile measurements are available.

$$N(h) = N(0)(ah^3 + bh^2 + ch + d), \qquad (2)$$

where $a$, $b$, $c$, and $d$ are the coefficients when $N(0) = 1$ (Russo et al., 2006)."

We hope you will agree that maintaining the exponential function alongside other options to provide users a comprehensive package, ensuring adaptability to diverse research contexts.

5. This section was jumpy and difficult to follow. I suggest further dividing into subsections:

**Reply:** In accordance with your suggestion, we have carefully divided section 3 into subsections as follows (Pages #10-18, Lines #196-348):

"

[revised manuscript text omitted]

"

6. Little mention of future work/applications. Maybe mention potential model development projects and associated improvement to probable mixing state predictions.

   **Reply:** The lines #486-490 (Page #25) are revised as;

   *"The prospective advancement linked to AeroMix involves the development of an optimization/machine learning algorithm utilizing AeroMix as the aerosol optical model. Such an algorithm will enable faster and more deterministic estimation of aerosol mixing states at fine temporal and spatial resolutions."*

**Minor comments**

**Line 43:** Healy et al. 2014 and Ye et al. 2018 describe in situ measurements of mixing state between external and internal.

**Reply:** The single-particle mass spectrometry observations presented in these studies will help present the case in our manuscript. Suggested references are incorporated to strengthen the discussion on real atmospheric aerosol mixing states. The lines #47-50 (Page #2) are revised as;

"Despite the experimental evidence that the aerosol mixing state lies between a purely externally mixed state and a purely internally mixed state (Healy et al., 2014; Li et al., 2016; Ye et al., 2018; Riemer et al., 2019), the standard FORTRAN-based OPAC model considers the external mixing of aerosols alone and cannot treat complex aerosol internal mixing states."

**Line 58:** Unclear to reader why "batch mode" would have limited functionalities. Either clarify or reword.

**Reply:** We removed this discussion about AEROgui to avoid ambiguity (Page #2, Line #63).

**Line 97-98:** Unclear why density of BC is mentioned here.

**Reply:** The specific densities for all components are sourced from the Global Aerosol Data Set and OPAC database. However, the observational studies on black carbon suggest using a specific density value of 1.8 g cm$^{-3}$, differing from the 1.0 g cm$^{-3}$ value provided in OPAC. The sentence is revised as follows (Page #5, Line#118):

"The density of BC is set at 1.8 g cm$^{-3}$ based on observations, which differs from the value of 1.0 g cm$^{-3}$ given in OPAC  (Bond et al., 2013; Kondo, 2015)."

**Line 137 – 139:** This sentence is confusing, specifically "and particles composed of multiple chemical species (internally mixed) as an external mixture."

**Reply:** The sentence (Page#5, line#94) is revised as follows for brevity.
"Optical properties of the complex aerosol mixing states are modeled by accounting for any number of both externally mixed particles (composed of single chemical species) and internally mixed particles (composed of multiple chemical species as core and shell), with no presumed chemical or physical interaction among the particles within the mixture (see Fig. 1)."

**Equation (7):** I believe index of summation should be layer, not n.

**Reply:** The equation is corrected as (Page #9, line# 191),
"

$$Total\ AOD = \sum_{n=1}^{6} \beta_{ext_n} \int_{h_{min_n}}^{h_{max_n}} N_n(h)dh$$

where $h_{min}$ and $h_{max}$ are the layer bottom and layer top height for each layer $n$."
**Line 163:**  Consider adding one sentence on the Mie inversion technique.

**Reply:** The sentence is revised as follows (Page #10, Line #198):

"The probable aerosol mixing states are modeled with AeroMix using the Mie inversion technique by iteratively comparing the multispectral measurements of aerosol optical properties with modeled ones for different mixing scenarios until they converge within the observational error (Fig. 1) (Chandra et al., 2004; Dey et al., 2008; Kaskaoutis et al., 2011; Ramachandran and Srivastava, 2016; Srivastava et al., 2016, Srivastava et al., 2018)"

**Figure 2:** Consider adding average wind directions for winter, pre-monsoon and post-monsoon. This would aid in the comparison of modeled mixing states discussion (Sections 4.3 and 4.4). Label IGP region.

**Reply:** Average wind directions for different seasons are added to Figure 2, along with labeling the IGP region as shown below (Page #14).

[Figure]

**Figure 2: a) Location of Kanpur (diamond) and the W-ICARB cruise track (dotted lines). Measurements of AOD, chemical composition, and relative humidity over the West-BoB and North-BoB (demarcated with dashed lines) were utilized in this study. Additionally, the locations of Delhi, Varanasi, and Bhubaneswar (circles) are indicated, where mixing state measurements were obtained for comparison with the modeled mixing states. The shaded area marks the Indo-Gangetic Plains. Seasonal average surface winds over the region from ERA5 reanalysis are shown for a) winter, b) premonsoon and c) postmonsoon by vectors.**

**Line 243:** Add reasoning for RMSE threshold after sentence: "Only those spectra with RMSE minimum 0.03 are considered as the best fit (Fig. 1). [Add here]." The last sentence of this paragraph "The RMSE threshold of 0.03 is chosen to ensure …" should be moved to this location.

**Reply:** Revised as suggested (Page #17, Line #311):

"Only those spectra with RMSE minimum ≤ 0.03 are considered as the best fit (Fig. 1). The RMSE threshold of 0.03 is chosen to ensure that the RMSE remains within the lowest uncertainty (15%) in the Aerosol Robotic Network (AERONET) AOD and SSA retrievals (Srivastava and Ramachandran, 2013)."

**Figure 3 and 4:** Unnecessary to include colors for externally mixed and core-shell mixed in caption.

**Reply:** Revised as suggested;

**Page #20**
"Figure 3: Measured and AeroMix modeled aerosol parameters over Kanpur during the winter, premonsoon, and postmonsoon seasons of 2007-2009. a) AeroMix modeled season-wise spectral AOD compared with AERONET measured AOD. Vertical bars represent the standard deviation of the mean values. Root mean square error of the fit is given in parenthesis, b) Same as (a) but for SSA, c) Component-wise aerosol mass concentration in externally mixed and core-shell mixed state compared with measured aerosol mass concentration. Vertical bars represent standard deviation of measurements, d) Modeled percentage mass fraction of aerosol components to the total aerosol mass (inner pie) and percentage mass fraction of externally mixed and core-shell mixed aerosols to the total aerosol mass (outer pie)."

**Page #22**
"Figure 4: Measured and AeroMix modeled aerosol parameters over W-BoB and N-BoB during winter season (December 2008-January 2009). a) AeroMix modeled season-wise spectral AOD compared with measured AOD. Vertical bars represent the standard deviation of the mean values. Root mean square error of the fit is given in parenthesis, b) Component-wise aerosol mass concentration in externally mixed and core-shell mixed state compared with measured aerosol mass concentration. Vertical bars represent standard deviation of measurements, c) Modeled percentage mass fraction of aerosol components to the total aerosol mass (inner pie) and percentage mass fraction of externally mixed and core-shell mixed aerosols to the total aerosol mass (outer pie)."

**Technical corrections**

**Line 11:** Add comma after "aerosol chemical compositions,"

**Reply:** Corrected as suggested (Page #1, Line #11):

"Assessing aerosol mixing states, which mainly depend on aerosol chemical compositions, is indispensable for estimating aerosol direct and indirect effects."

**Line 13:** Remove "the" so the sentence reads "While limitations in the measurements of aerosol …"

**Reply:** The sentence is modified as per the next comment.

**Line 13-15:** Sentence "While the limitations in the measurements …" is a run on. Consider breaking into two sentences.

**Reply:** Following your suggestion, we corrected the sentence as (Page #1, Line #12):

"The limitations in the direct measurements of aerosol chemical composition and mixing states necessitate modeling approaches to infer the aerosol mixing states. The Optical Properties of Aerosols and Clouds (OPAC) model has been extensively utilized to construct optically equivalent aerosol chemical compositions from measured aerosol optical properties using Mie inversion."

**Line 18:** Move position of "3)" – "components, and 3) define aerosol composition"

**Reply:** Corrected as suggested (Page #1, Line #19):

"2) simulate optical properties of aerosol mixtures constituted by any number of aerosol components, and 3) define aerosol composition and relative humidity in up to six vertical layers."

**Line 98:** Add comma "The optical properties of each component, except for the MD components, are calculated"

**Reply:** Corrected as suggested (Page 5, Lines #119-121):

"The optical properties of each component, except for MDnm, MDam, and MDcm components, are calculated using the Mie theory (Mie, 1908)."

**Line 209:** Run-on sentence, rewrite as "…CSR values of core-shell mixed aerosols in the model. This offers a more flexible approach compared to relying on measured…"

**Reply:** Corrected as suggested (Page #15, Line #261):

"In contrast, this study proposes allowing the variations of CSR values of core-shell mixed aerosols in the model. This offers a more flexible approach than relying on measured $M_a$ to assess the probable aerosol mixing state."

**Line 297:** Change to "… mixed states varied from 30% to 59% in premonsoon and winter, **respectively.**"

**Reply:** Changing to "mixed states varied from 30% to 59% in premonsoon and winter, respectively" may lead to confusion regarding whether these values represent intra-seasonal or inter-seasonal variation. To clarify, we have revised the line #374 (Page #21):

"The percentage mass fractions of the total $M_a$ in core-shell mixed states increased from 30% in premonsoon to 59% in winter (Fig. 3d)."

**This manuscript is generally well-composed.**

**Reply:** We appreciate and thank the reviewer for his/her constructive comment and recommendation. We have incorporated the necessary corrections in the revised manuscript as described below.

**Dividing the content between lines 80 to 120 into more succinct paragraphs could significantly improve readability.**

**Reply:** Following your suggestion, we have divided section 2 into two subsections and added a table to explain pre-defined components to improve readability. Section 2 is restructured and revised as follows (Pages #3-10, Lines #80-195).
"

[revised manuscript text omitted]

**Reviewer: # 3**

Overall the paper is written and presented very well, though I agree with the other referees about rearranging section 3 to make a smoother read.
Verdict: Accept with minor revisions (detailed below)

**Reply:** We appreciate and thank the reviewer's insightful and detailed comments and recommendation. All comments have been taken into account, as described below.
We have also restructured sections 2 and 3 following the first and second reviewer's comments.

**General Comments**

1. In the introduction (e.g. line 53), the limit on number of aerosol components that are available in OPAC is presented as a key limitation on its ability to infer mixing state. However, on line 103, the authors explain how AeroMix relies on the OPAC aerosol database for its measurements. My understanding is that AeroMix is therefore limited to the number of components available in the OPAC aerosol database. Therefore, whilst both OPAC and AeroMix rely on the OPAC aerosol database, are they not identically limited to the number of aerosol components? (By the way, I note there is an argument that users could append aerosol components to the OPAC aerosol database, and therefore extend the number of aerosol components present in AeroMix, but is this not equally true for the OPAC model? If not, then please explain.)

**Reply:** While both OPAC and AeroMix use the OPAC aerosol database, the limitation of OPAC is not due to the number of components available in the database (which are nine). It is due to the maximum number of components permissible in a mixture is seven for calculating the optical properties. This is specified in the OPAC input file as depicted below.

> *# if you did select 0 (define new mixture), in the next 5 lines the*
> *# component number and the number density in [particles/cm\*\*3] has to be*
> *# given, divided by commas. The component numbers are the following:*
> *#*
> ***# There are not more than 7 components allowed to compose one aerosol***
> ***# or cloud type.***
> *#*
> *# (1) insoluble          (6) mineral (nuclei mode, nonspherical)*
> *# (2) water-soluble        (7) mineral (accumulation mode, nonspherical)*
> *# (3) soot               (8) mineral (coarse mode, nonspherical)*
> *# (4) sea-salt (accumulation mode)*
> *# (5) sea-salt (coarse mode) (10) sulfate*

In contrast, AeroMix offers more flexibility, which can account for any number of aerosol components to constitute a mixture. It does utilize all the nine predefined components available in the OPAC database. In

addition, it also allows user-defined aerosols in the mixture as explained in lines #124-127. In this study, along with the predefined components, their core-shell mixed combinations at various mass mixing ratios are also considered, totaling 305 components as explained in line #247.

Adding new aerosol components to OPAC would require compromising the predefined components due to the program's limitation to account for more than seven components in a mixture. With the capability of AeroMix to model any number of aerosol components in a mixture, this study assessed the probable coexistence of aerosol components in externally mixed states and different core-shell mixed states at various mass-mixing scenarios, considering all possibilities simultaneously, which is why we believe it offers a more comprehensive approach to assessing the aerosol mixing state as explained in line #387.

2. It is clear from Fig. 1 that AeroMix, as a stand-alone model predicts the optical properties of aerosol populations, and I can see from line 73 and line 98 that (as in OPAC) Mie theory is used to do this, and this is represented by the 'AeroMix' item in Figure 1. However, I am confused by the Mie inversion part of AeroMix: in line 163 several references are given for the 'Mie inversion' technique to find the aerosol mixing state, however none of these references use the phrase 'Mie inversion' for their techniques. Is this phrase newly coined in this paper? If so, it should be properly justified, including an explanation of why 'inversion' is the correct word, rather than iteration. I say this because it appears from Fig 1 and the description in line 241-260 that an iterative approach is used to find the mixing state that gives best fit to observations. Therefore, I question whether this is an inversion approach, or an iterative approach. Would 'Mie iteration' be a better phrase?

**Reply:** We acknowledge that the term 'Mie inversion' is not used in any of the cited references. However, related phrases such as 'inverting Mie model' or 'inverse Mie problem' are frequently used when estimating aerosol properties like refractive indices from optical properties. For example, Sumlin et al. (2018) discuss solving the 'inverse Mie problem' for a complex refractive index given inputs of scattering, absorption, and size distribution parameters. Similarly, Pedrós et al. (2014) refer to 'inverting the Mie model' to infer aerosol properties from measured optical properties.

The forward Mie calculation process involves determining scattering and absorption parameters at specific wavelengths by inputting the radius and complex refractive index of the particle. Consequently, estimating particle properties, in this case, the aerosol component characterized by its spectral refractive indices and size distribution parameters, from measured optical properties, presents an inverse Mie problem. The solution to this problem is obtained by iteratively finding a combination of aerosol components (an aerosol mixture) that yields aerosol optical depth and single scattering albedo spectra that best match the measured ones, with a minimum root mean square error (RMSE) between the modeled and measured spectra (as explained in line #309).

Given the context above, we believe the term 'Mie inversion' is more appropriate than 'Mie iteration', as it accurately reflects the process of solving the inverse problem, even though it involves iterative steps.

3. Line 241 indicates that the iteration in AeroMix finds the probable existence of components, but Fig 1 and other part of the main text (e.g. line 253) suggest the iteration finds the probable mixing state. What does the iteration solve? Is it both the components present and their mixing state? If so, please make clearer.

**Reply:** The iteration specifically solves for the number concentration of aerosol components, with non-zero values indicating the probable existence of these components which refers to the aerosol mixing state. Line #309 (Page #17) is revised as follows for clarity:

"The probable existence of the aerosol components in the atmosphere, which refers to the aerosol mixing state is assessed by iteratively varying the number concentrations in the mixture in AeroMix until the root mean squared error (RMSE) between the measured and modeled AOD and SSA spectra are minimized."

4. The authors acknowledge that the iterative technique used to estimate probable mixing states is not unique (line 254). I would like to see more discussion, or reference (where this issue has been detailed before), around the range of mixing states that can be inferred from the iteration technique for a given set of inputs, i.e., a discussion around the probability that a given inferred mixing state is accurate. If I understand that AeroMix returns just one probable mixing state for a given set of inputs, then this seems to me to be a fundamental weakness, and future work should prioritise quantifying the uncertainty around the returned mixing state, and/or returning multiple mixing states so that users can quantify the range of probable solutions. I see that this issue is dealt with in lines 404-407, which explains that other algorithms could imbed AeroMix and therefore quantify the probable accuracy of returned mixing states. I also see that in sections 4.4 and 5, that the authors express the limitations of AeroMix, stating that only results useful for qualitative interpretation are returned. This is great, because it's key that the authors describe the model's limitations. However, I think this limitation should be mentioned in the model overview, so that readers are quickly aware of it. Furthermore, it would be useful to have some more description around what the 'inherent constraints' mentioned in line 401 are – this would greatly help authors of future minimization and machine learning algorithms to correctly utilise AeroMix.

**Reply:** The above reviewer comment has been segmented into four distinct sections for a more structured response:

a) The authors acknowledge that the iterative technique used to estimate probable mixing states is not unique (line 254). I would like to see more discussion, or reference (where this issue has been detailed before), around the range of mixing states that can be inferred from the iteration technique for a given set of inputs, i.e., a discussion around the probability that a given inferred mixing state is accurate.

We have mentioned in line #328 that the aerosol mixing state modeled by the Mie inversion technique is not unique, but a probable one. When solving a system of linear equations where the number of unknowns (number concentration of aerosol components) is greater than the number of constraints (spectral AODs and SSAs), multiple solutions are possible. This is because there are insufficient constraints to uniquely determine all the unknowns. While many possible solutions exist mathematically, not all are physically feasible for aerosols. To select a physically feasible solution for the location under consideration, we further constrained the modeled mixing states with concurrent and collocated measurements of component-wise aerosol mass concentration (Page #17, Line #332). The solutions in which the total aerosol mass concentration of each component (externally and core-shell mixed mass combined as explained in lines #267-275) within the bounds of $\pm 1\sigma$ of the measured component-wise aerosol mass concentration is considered acceptable.

We have revised line #328 (Page #17) to clarify this point as follows.

*"It is important to note that the aerosol mixing state modeled by the Mie inversion technique is not unique but a probable scenario. Solving a system of equations with number of unknowns greater than the constraints poses an undetermined system having multiple solutions (Sumlin et al., 2018). Hence, a physically feasible solution is selected from the AeroMix-modeled probable aerosol mixing states by further constraining with the measured component-wise $M_a$ with the modeled $M_a$ agreed to within $\pm 1\sigma$."*

b) If I understand that AeroMix returns just one probable mixing state for a given set of inputs, then this seems to me to be a fundamental weakness, and future work should prioritise quantifying the uncertainty around the returned mixing state, and/or returning multiple mixing states so that users can quantify the range of probable solutions. I see that this issue is dealt with in lines 404-407, which explains that other algorithms could imbed AeroMix and therefore quantify the probable accuracy of returned mixing states.

As explained above, AeroMix provides not just one solution for a given set of inputs, but different probable ones depending on the combinations examined. The key limitation of this technique is that it does not guarantee a unique solution but a probable one. To assess the probability of a particular solution being true or to quantify the range of uncertainty of the solution, all possible solutions need to be modeled. This is not practical considering the number of combinations to be examined and the required computational time. For example, if we check 1000 possible values for all 305 components, the total iterations required will be $1000^{305}$. As the reviewer suggested for future work, the development of minimization and machine learning algorithms is already underway to ensure a quantitative and deterministic estimation of aerosol mixing states using AeroMix as discussed in lines #486-490 (Page #25).

c) I also see that in sections 4.4 and 5, that the authors express the limitations of AeroMix, stating that only results useful for qualitative interpretation are returned. This is great, because it's key that the authors describe the model's limitations. However, I think this limitation should be mentioned in the model overview, so that readers are quickly aware of it.

The limitation of the AeroMix in modeling aerosol mixing states using the Mie inversion technique is that the modeled mixing states are not unique, but probable scenarios. This inherently limits the analysis of mixing states using AeroMix to a qualitative examination. Lines #83-86 (Page #3) in the model overview section are revised as follows to indicate that the modeled mixing states are probable ones.

"The workflow of AeroMix for modeling the aerosol properties and assessing the probable mixing states is illustrated in Fig. 1. A methodology for modeling the probable aerosol mixing state using AeroMix is detailed in the subsequent section."

d)  Furthermore, it would be useful to have some more description around what the 'inherent constraints' mentioned in line 401 are – this would greatly help authors of future minimization and machine learning algorithms to correctly utilise AeroMix.

The inherent limitations of the Mie inversion technique are detailed in response to the first part of this comment (a) as well as in section 3.3 in the revised manuscript. Lines #481-483 in Section 5 are revised to reflect this.

"However, this study is limited to a qualitative examination of aerosol mixing states due to the inherent constraints of the inverted Mie model approach as discussed in Sect. 3.3."

5. Please detail what happens along the 'No' route of Fig. 1 – the authors explain in line 241 that iterative changes are made, but how is the amount and direction of change estimated?

**Reply:** The direction of change is decided towards where the root mean square error between the modeled and measured AOD and SSA spectra are converged as mentioned in lines #309-311 (Page #17). Determination of the amount of change of component number concentration is under development and will be addressed in future work.

**Minor Comments**

Line 35 – please provide some example references of studies that have used OPAC to estimate probable mixing state.

**Reply:** The references cited in lines #49 and #163 have used OPAC to estimate the probable mixing state.

Line 57 – more detail is needed about why AEROgui is inferior to AeroMix, so that readers can distinguish between the two. For example, what limits in functionality does AEROgui have that AeroMix overcomes?

**Reply:** We removed this discussion about AEROgui to avoid ambiguity following the first reviewer's comment.

Line 137 – I don't understand the sentence starting on this line. Specifically, how can an internally mixed aerosol be treated as an external mixture? This sounds like the components comprising the internally mixed aerosol are separated into separate particles so that an external mixture is formed.

**Reply:** To clarify, for the Mie inversion technique, all the probable aerosol components need to be modeled in AeroMix beforehand to examine their existence in the mixture. Core-shell mixed aerosols are modeled as user-defined aerosol components, as explained in section 2.1.2 (Page #8) before they are constituted into a mixture. AeroMix treats this core-shell mixed aerosol in the same manner as an externally mixed aerosol. In other words,

for AeroMix, a core-shell mixed aerosol is another aerosol with characteristic spectral Mie coefficients, size distribution parameters, and specific density ($\rho$). However, these Mie coefficients are calculated using the coated-sphere Mie scattering theory and specific density as explained in line #158. AeroMix assumes no chemical or physical interaction among the particles within the mixture while calculating the optical properties.

Subsequently, different combinations of predefined externally mixed aerosols (composed of single chemical species) and internally (core-shell) mixed aerosols (composed of multiple chemical species) are used to constitute various aerosol mixtures in AeroMix to calculate their optical properties.

The sentence (Page #5, Lines #94-96) is revised as follows for brevity.

*"Optical properties of the complex aerosol mixing states are modeled by accounting for any number of both externally mixed particles (composed of single chemical species) and internally mixed particles (composed of multiple chemical species as core and shell), with no presumed chemical or physical interaction among the particles within the mixture (see Fig. 1)."*

---

## Referee Report (RR1)

I think that Raj et al have responded well to the reviewers' comments and have produced an article that is of high quality and certainly within the scope of GMD. Therefore I recommend publication as is.

---

## Author Response (AR2)

**Cover letter**

To

Dr. Sylwester Arabas
Topic Editor,
Geoscientific Model Development (GMD)

02 July 2024

**Subject: Author reply to the Editor's comments**
Manuscript Title: AeroMix v1.0.1: a Python package for modeling aerosol optical properties and mixing states.
Manuscript ID: EGUSPHERE-2024-62

Dear Dr. Sylwester Arabas,

Along with this letter, we submit the revised version of our manuscript titled "AeroMix v1.0.1: A Python package for modeling aerosol optical properties and mixing states" to the Journal of Geoscientific Model Development (GMD). We are very glad to note that all three referees have recommended our manuscript for acceptance.

The co-authors and I appreciate your comments on ensuring the proper formatting of the manuscript according to GMD policies and enhancing the reproducibility of the results. We have carefully considered your feedback and made the necessary revisions to the manuscript file and the scripts. Our responses to your comments are attached, with line numbers referenced as they appear in the tracked changed version.

All the authors agreed with the revision of this manuscript. I request you to kindly take the necessary steps for the manuscript's earliest decision.

Thank you.

Sincerely yours,

P. R. Sinha
Department of Earth and Space Sciences
Indian Institute of Space Science and Technology (IIST)
Valiamala, P.O., Thiruvananthapuram-695 547, India
Phone No. (Off): +91-471-2568523
e-mail id: prs@iist.ac.in

**Response notes**

1. Note that GMD reserves supplement for items that cannot reasonably be included in the main text or as appendices (https://www.geoscientific-model-development.net/submission.html#assets). Supplements are also not subject to language editing, reference checks and typesetting in the publication process. Please replace the supplementary material pdf with appendices to the paper - to comply with the policy and to ensure it is consistently typeset and edited with the main body of the paper (currently, the title does not match the paper with lack of version information and "state" vs. "states" mismatch)

   **Reply:** In accordance with GMD policy, the supplementary material has been merged into the manuscript as Appendices A, B, and C. We request the removal of the previously submitted supplementary material.

2. The reference "Ram et al. (2010)" on page 3 in the supplement is ambiguous - please clarify which paper is cited - 2010a or 2010b?

   **Reply:** Corrected as suggested in Appendix B.1 (line #465).
   "A detailed description of the analytical procedures and the data can be found in Ram et al. (2010a)."

3. There are hardcoded absolute paths ("/home/sam/CommonFiles/PhD") in the Python script provided - please ensure that all scripts can be run by the readers without errors, in other environments than author's machine - use relative paths

   **Reply:** Absolute paths are now replaced with relative paths in the script, and the updated scripts have been uploaded to Zenodo (https://doi.org/10.5281/zenodo.12625421). The code availability section has been updated to reflect this change (line #533).
   "The model outputs and codes used to generate the results presented are available in a separate Zenodo repository: https://doi.org/10.5281/zenodo.12625421 (P Raj and Sinha, 2024b)."

4. The Zenodo link leads to an archive consisting of numerous .ods, .xlsx, .inp, .out and .png files + three Python scripts, but without any explanation on the contents or usage of these files. The Python scripts do not refer to any of the .ods files. Please include an explanation on the contents of these files. How a reader should proceed with using these files and with what kind of software? The dependency on scikit-learn and openpyxl should also be explained - either in the text, a README file or a requirements.txt list of dependencies. I highly recommend to check if the complete set of figures can be recreated by someone else on a different machine before resubmission to Zenodo. Same applies to regenerating the output data from OPAC and AeroMix.

The updated archive, which includes the scripts and spreadsheets (.ods, .xlsx) used for data analysis and visualization of each figure, has been uploaded to Zenodo. The manuscript provides the link to this archive. Each folder now contains a README file explaining the script requirements and the contents of each file. Due to the large volume of source data, only the processed data necessary for producing the figures are included. Source datasets can be accessed from the respective sources listed in the data availability section.

5. Only one of the supplied Python scripts uses AeroMix package. Scripts for Fig 3 and Fig 4 have all the AeroMix output data manually entered into the code. Please include scripts used to setup and run AeroMix to achieve these output data.

   **Reply:** We have included the scripts used to set up and run AeroMix, along with the visualization scripts.